# COMAL: A CONVERGENT META ALGORITHM FOR ALIGNING LLMS WITH GENERAL PREFERENCES

**Yixin Liu**[*1], **Argyris Oikonomou**,[*] **Weiqiang Zheng**[*1], **Yang Cai**[†1], **Arman Cohan**[†1,2]
[1]Yale University, [2]Allen Institute for AI
{yixin.liu, argyris.oikonomou, weiqiang.zheng}@yale.edu
{yang.cai, arman.cohan}@yale.edu

## ABSTRACT

Many alignment methods, including reinforcement learning from human feedback (RLHF), rely on the Bradley-Terry reward assumption, which is not always sufficient to capture the full range and complexity of general human preferences. We explore RLHF under a general preference framework by modeling the alignment problem as a two-player zero-sum game in a game-theoretic framework, where the Nash equilibrium policy guarantees a 50% win rate against any competing policy. However, previous self-play algorithms for finding the Nash policy either diverge or only converge to a Nash policy in a modified game, even in a simple synthetic setting, thereby failing to maintain the 50% win rate guarantee against all other policies. We propose a meta-algorithm, **Co**nvergent **M**eta **Al**ignment Algorithm (COMAL), for language model alignment with general preferences, inspired by convergent algorithms in game theory. We provide theoretical analysis that our meta-algorithm converges to an exact Nash policy in the last iterate and demonstrate its effectiveness on a range of synthetic and preference optimization datasets. COMAL is simple and can be integrated with many existing methods designed for preference optimization with minimal changes, and empirically it consistently maintains above 60.2% and 56.8% win rates, when applied to Llama-3-8B-Instruct and Qwen2.5-7B, against all compared algorithms under controlled evaluations.

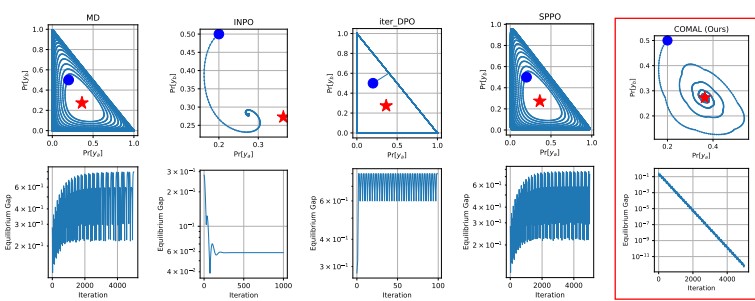 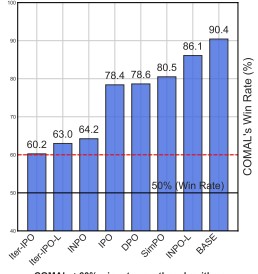

(a) COMAL converges to the optimal solution, while other preference optimization methods do not. We initialize all algorithms at the blue dot; the Nash equilibrium is the red star.

(b) COMAL's win rate against other PO algorithms.

Figure 1: (a) convergence behavior of five methods (§4); (b) win-rate comparison with Llama-3 (§5).

## 1 INTRODUCTION

One of the most widely adopted approaches to addressing the challenge of aligning LLMs with human values and preferences is Reinforcement Learning from Human Feedback (RLHF) (Christiano et al., 2017; Ouyang et al., 2022). This framework consists of two steps: first, learning a reward

---

[*]Equal contribution; alphabetically ordered.

[†]Equal co-advising; alphabetically ordered.

model from a human preferences dataset, and second, optimizing the LLM using the proximal policy optimization (PPO) algorithm (Schulman et al., 2017). More recently, Rafailov et al. (2024) observed that the first step can be bypassed, proposing the direct preference optimization (DPO) algorithm, which directly optimizes the LLM from the dataset.

However, the aforementioned approaches crucially rely on the assumption that human preferences can be expressed using the Bradley-Terry (BT) model (Bradley & Terry, 1952). Unfortunately, the BT model is too restrictive to capture the richness and complexity of human preferences. For example, the BT model can only induce *transitive* preferences – i.e., if more people favor A over B, and B over C, then more people must favor A over C. Such transitivity may not hold in the presence of diverse populations and is also incompatible with evidence from human decision-making (May, 1954; Tversky, 1969). To illustrate this, consider a simple case where users are evaluating responses from an assistant to a nuanced question like: "What's the best way to spend a Sunday?" Some might prefer Response A (outdoor activities) over B (reading a book), while others prefer B over C (watching TV), yet a third group prefers C over A. These cyclic preferences – A > B > C > A – cannot be modeled by the BT framework. Moreover, even if each individual has a consistent (transitive) ranking, the aggregated preferences can exhibit intransitivity. In fact, even a mixture of two BT models cannot be parameterized by a single BT model.

To overcome this limitation, recent research has begun to explore alignment under general preferences. Munos et al. (2024); Swamy et al. (2024) formulate this alignment problem as a symmetric two-player zero-sum *alignment game* (Definition 2), where both players' strategies are LLMs, and their payoffs are determined by the win rate against the opponent's LLM according to the preference model. The objective is to identify a Nash equilibrium policy, which guarantees at least 50% win rate against any competing policy. Existing algorithms for finding such robust policies present significant challenges. In particular, current methods can suffer from instability, often failing to converge, or may inadvertently optimize for a solution to a modified version of the original problem. As a result, these approaches may fail to guarantee the desired win rate against arbitrary opponents, leaving robust alignment an open and active area of research and motivating us to investigate the following question:

**Question:** Is there an algorithm that *converges* to the Nash equilibrium policy of the alignment game (Definition 2), thus guaranteeing 50% win-rate against any competing policy?

**Our Contributions:** We propose a novel meta-algorithm, the **Co**nvergent **M**eta **Al**ignment Algorithm (COMAL), that iteratively refines language model policies by solving a regularized two-player zero-sum game at each round, using the current policy as a reference point. The procedure at each round is as follows:

*Step 1:* In iteration $t$, solve a KL-regularized two-player zero-sum game with respect to the reference policy $\pi_{\text{ref}} = \pi_{t-1}$. Let $\pi_t$ be the Nash equilibrium of this regularized game.

*Step 2:* Update the reference policy $\pi_{\text{ref}}$ to the current policy $\pi_t$ and repeat the process.

The rationale behind COMAL is that it is a practical implementation of the Conceptual Prox-method (Nemirovski, 2004), a *convergent* algorithm for solving two-player zero-sum games, whether regularized or not. Importantly, Step 1 can be implemented using the Prox operator, a well-known concept in the optimization literature (Parikh et al., 2014). A crucial observation we make here is that many existing algorithms – including PPO (Schulman et al., 2017), GRPO (Shao et al., 2024; Guo et al., 2025), DPO (Rafailov et al., 2024), IPO (Azar et al., 2024), SPPO (Wu et al., 2024), REBEL (Gao et al., 2024), DRO (Richemond et al., 2024), and INPO (Zhang et al., 2025b), *inter alia* – can be interpreted as practical implementations of the Prox operator in the context of LLM training (see §3.3 for a detailed discussion). As a result, COMAL is simple and can be integrated with many existing methods designed for preference optimization with minimal changes.

One significant departure of COMAL from existing methods for the game-theoretic formulation of alignment is that we adaptively update the reference policy rather than keeping it fixed. A potential concern with this approach is that the policy might drift too far from the initial policy, leading to instability and quality degradation. However, we provide both theoretical guarantees and experimental evidence demonstrating that this dynamic updating strategy consistently *enhances* model performance while maintaining stability.

**Theoretical guarantee:** Given any implementation of the Prox operator, COMAL provably converges to the Nash equilibrium policy in the last iterate. While existing algorithms like iterative

IPO (Azar et al., 2024) and SPPO (Wu et al., 2024) only guarantee average-iterate convergence (which is impractical for LLMs) or convergence to a KL-regularized Nash equilibrium (Munos et al., 2024; Zhang et al., 2025b), COMAL is the *first* algorithm that has provable last-iterate convergence to the unregularized Nash equilibrium.[1]

**Empirical improvements:** We first conduct synthetic controlled experiments on a $3 \times 3$ two-player zero-sum alignment game and demonstrate that COMAL is the only algorithm that converges to the Nash equilibrium. Under realistic LLM training settings, in experiments with Llama-3-8B-Instruct (Dubey et al., 2024) and Qwen2.5-7B (Yang et al., 2024b) on UltraFeedback (Cui et al., 2023), COMAL achieves above 60% and 56% win rates, respectively, against all compared algorithms according to the preference oracle.

## 2 BACKGROUND

We begin by introducing notation for language model alignment and preference modeling. Let $\Delta(\mathcal{Z})$ denote the set of distributions over a set $\mathcal{Z}$. Let $\mathcal{X}$ be the instruction set with a fixed distribution $\rho \in \Delta(\mathcal{X})$, and $\mathcal{Y}$ be the response set. Given an instruction $x \in \mathcal{X}$, an LLM policy $\pi$ specifies an output distribution $\pi(\cdot \mid x) \in \Delta(\mathcal{Y})$. For $p, q \in \Delta(\mathcal{Z})$, the Kullback-Leibler (KL) divergence is $\mathrm{KL}(p\|q) := \sum_{z \in \mathcal{Z}} p(z) \log \frac{p(z)}{q(z)}$. The sigmoid function is $\sigma(x) := \frac{e^x}{e^x+1}$. We use $\mathrm{supp}(p)$ to denote the support of distribution $p$. This paper focuses on general preference models.

**Definition 1** (General Preference Model). *A general preference model* $\mathbb{P} : \mathcal{X} \times \mathcal{Y} \times \mathcal{Y} \to [0, 1]$ *satisfies* $\mathbb{P}(y_1 \succ y_2 \mid x) = 1 - \mathbb{P}(y_2 \succ y_1 \mid x)$. *When we query* $\mathbb{P}$ *with* $(x, y_1, y_2)$, *it outputs* 1 *with probability* $\mathbb{P}(y_1 \succ y_2 \mid x)$ *meaning* $y_1$ *is preferred over* $y_2$, *and it outputs* 0 *otherwise. The* win rate *of* $\pi_1$ *over* $\pi_2$ *under preference model* $\mathbb{P}$ *is* $\mathbb{P}(\pi_1 \succ \pi_2) := \mathbb{E}_{x \sim \rho}[\mathbb{E}_{y_1 \sim \pi_1, y_2 \sim \pi_2}[\mathbb{P}(y_1 \succ y_2 \mid x)]]$.

We present the Bradley-Terry (BT) model and additional backgrounds on RLHF and DPO to §B.

### 2.1 ALIGNMENT WITH GENERAL PREFERENCE MODELS

The Bradley-Terry (BT) model, while widely used in preference modeling, has fundamental limitations that restrict its ability to capture the full complexity of human preferences such as intransitive preferences, especially when aggregating preferences across diverse populations or when dealing with nuanced, context-dependent decisions (Munos et al., 2024; Swamy et al., 2024). To address these limitations and achieve alignment with general preferences, following (Munos et al., 2024; Swamy et al., 2024), we model the policy optimization problem as a two-player zero-sum game.

**Definition 2** (Alignment Game). *The* alignment game *is a two-player zero-sum game with objective*

$$J(\pi_1, \pi_2) := \mathbb{P}(\pi_1 \succ \pi_2) - \frac{1}{2}. \tag{1}$$

*The constant* $\frac{1}{2}$ *is introduced only to ensure the game is zero-sum and it has no other effect. We focus on policies with* $\Pi := \{\pi : \mathrm{supp}(\pi) \subseteq \mathrm{supp}(\pi_{\mathrm{init}})\}$ *in the support of the initial policy. A* ***Nash equilibrium*** *policy is* $(\pi_1^\star, \pi_2^\star) \in \mathrm{argmax}_{\pi_1 \in \Pi} \mathrm{argmin}_{\pi_2 \in \Pi} J(\pi_1, \pi_2)$ *and satisfies* $J(\pi_1, \pi_2^\star) \leq J(\pi_1^\star, \pi_2^\star) \leq J(\pi_1^\star, \pi_2), \forall \pi_1, \pi_2 \in \Pi$.

In this game, the max-player controls $\pi_1$ and tries to maximize $J(\pi_1, \pi_2)$ while the min-player controls $\pi_2$ and tries to minimize $J(\pi_1, \pi_2)$. Since the game for two players $J(\pi_1, \pi_2)$ is symmetric (Ye et al., 2024), the game has a symmetric Nash equilibrium $(\pi^\star, \pi^\star)$. Moreover, the Nash equilibrium policy $\pi^\star$ guarantees that for any other policy $\pi$, its win rate is at least $\mathbb{P}(\pi^\star \succ \pi) \geq \mathbb{P}(\pi^\star \succ \pi^\star) = 50\%$. Our goal is to find a Nash equilibrium policy.

---

[1]We remark that a concurrent work (Wang et al., 2025) proposes an algorithm based on Magnetic Mirror Descent (Sokota et al., 2023) with last-iterate convergence. Our algorithms and theirs are all variants of the conceptual prox algorithm (Nemirovski, 2004). While their theoretical results require solving a regularized game *exactly*, we provide stronger results showing last-iterate convergence under the more practical setting with only *approximate* solutions (see Theorem 3). Their experiments and our experiments both confirm the effectiveness of convergent regularized learning algorithms for LLM alignment. We include a more detailed comparison with (Wang et al., 2025) in Appendix A

Existing online iterative preference optimization methods designed for or applicable to the original game, including iterative IPO (Azar et al., 2024) and SPPO (Wu et al., 2024), are based on Multiplicative Weights Update (MWU, definition in §3.2), and thus *diverge in the last iterate* as we show in §4.[2] There is also a line of works including Nash-MD (Munos et al., 2024; Ye et al., 2024), Online IPO (Calandriello et al., 2024), INPO (Zhang et al., 2025b) aim to find the Nash equilibrium of a modified KL-regularized game $J_\tau(\pi_1, \pi_2, \pi_{\text{ref}})$ defined as

$$J(\pi_1, \pi_2) - \tau \mathbb{E}_{x\sim\rho}[\text{KL}(\pi_1(\cdot \mid x)||\pi_{\text{ref}}(\cdot \mid x))] + \tau \mathbb{E}_{x\sim\rho}[\text{KL}(\pi_2(\cdot \mid x)||\pi_{\text{ref}}(\cdot \mid x))]. \quad (2)$$

The additional KL regularization terms in the objective are introduced for training stability. However, the Nash equilibrium of the modified game no longer guarantees a win rate of at least 50% against any competing policy. We compare these algorithms in Table 4.

Moreover, most existing theoretical convergence guarantees only hold for the average iterate, i.e., the uniform mixture of training iterates, which is not used in practice. The last iterate is widely used in practice, is more space-efficient (Munos et al., 2024), and has better performance demonstrated by existing experimental results (Munos et al., 2024; Wu et al., 2024; Zhang et al., 2025b). This motivates us to design principled algorithms with provable last-iterate convergence to Nash equilibrium policy.

## 3 CONVERGENT META-ALGORITHM FOR ALIGNMENT

We propose a simple meta-algorithm, **Co**nvergent **M**eta **Al**ignment Algorithm (COMAL, Algorithm 1), for aligning LLMs with general preferences. In §3.1 and 3.2, we present the theoretical foundations of COMAL and analyze its convergence properties. §3.3 describes its practical implementation that integrates COMAL with existing preference learning methods.

### 3.1 COMAL

We now introduce COMAL, our meta-algorithm for preference-based policy optimization, inspired by the conceptual prox-method (Nemirovski, 2004) from convex optimization and game theory. The prox-method has recently demonstrated strong practical performance in computing Nash equilibria for large-scale two-player zero-sum games (Perolat et al., 2021; Song et al., 2020; Abe et al., 2024) and has proven highly effective for the training of advanced game-theoretic AI systems (Perolat et al., 2022). Here, we adapt this framework into an online iterative procedure that guarantees convergence to the Nash equilibrium in the alignment game $J(\pi_1, \pi_2)$ (1).

---

**Algorithm 1: Co**nvergent **M**eta **Al**ignment Algorithm (COMAL) for solving alignment game

---

**Input:** Initial policy $\pi_{\text{init}}$, preference oracle $\mathbb{P}$, regularization $\tau > 0$, number of iterations $T \geq 1$
**Output:** Optimized policy $\pi^T$
Initialize $\pi^1, \pi_{\text{ref}} \leftarrow \pi_{\text{init}}$
**for** $t = 1, 2, \ldots, T - 1$ **do**
$\quad$ $\pi^{t+1} \leftarrow \text{argmax}_{\pi_1} \min_{\pi_2} J_\tau(\pi_1, \pi_2, \pi_{\text{ref}})$ using Algorithm 2 (discussed in §3.2)
$\quad$ $\pi_{\text{ref}} \leftarrow \pi^{t+1}$
**return** $\pi^T$

---

**Algorithmic Structure and Motivation.** At each iteration $t$, COMAL formulates and solves a *regularized zero-sum game*, defined by the objective $J_\tau(\pi_1, \pi_2, \pi_{\text{ref}})$ (2), where the regularization encourages policies to remain close (in KL divergence) to a reference policy $\pi_{\text{ref}}$. Specifically, the next policy $\pi^{t+1}$ is identified as a Nash equilibrium of this regularized game, with the current reference set to $\pi_{\text{ref}} = \pi^t$. (See Algorithm 2 and further exposition in §3.2). After convergence within this regularized subproblem, the reference policy is *updated* to the newly computed $\pi^{t+1}$ (the latest iterate): $\pi_{\text{ref}} \leftarrow \pi^{t+1}$, and the process repeats. This mechanism operationalizes a central insight of proximal algorithms: by updating the regularization center only when a regularized Nash equilibrium is reached, we ensure stable yet progressive movement toward the Nash equilibrium.

**Convergence and Monotonicity Guarantee.** A key property of COMAL is that the KL divergence to the Nash equilibrium policy $\pi^\star$ of the orginal game is monotonically non-increasing:$\text{KL}(\pi^\star||\pi^{t+1}) \leq$

---

[2]The MWU algorithm only has a weaker average-iterate convergence, i.e., $\frac{1}{T}\sum_{t=1}^{T}\pi^t$ converges.

$\mathrm{KL}(\pi^\star\|\pi^t)$. This holds for any choice of $\tau > 0$ (Lemma 4), permitting the regularization strength to be adaptively adjusted during training without requiring a vanishing decay schedule. Each iteration thus provably brings the policy closer to the original Nash solution, justifying the update of the reference policy.

**Theorem 1.** *We assume that there exists a Nash equilibrium $\pi^\star$ of $J(\pi_1, \pi_2)$ (defined in (1)) such that $\mathrm{supp}(\pi^\star) = \mathrm{supp}(\pi_{\mathrm{init}})$. In every iteration $t \geq 1$, it holds that $\mathrm{KL}(\pi^\star\|\pi^{t+1}) \leq \mathrm{KL}(\pi^\star\|\pi^t)$. Moreover, COMAL has last-iterate convergence, i.e., $\lim_{t\to\infty} \pi^t$ exists and is a Nash equilibrium.*

Moreover, while prior works (Perolat et al., 2021; Abe et al., 2024; Wang et al., 2025) require each regularized game to be solved *exactly*, we prove a stronger result (Theorem 3): last-iterate convergence holds even when each regularized game is solved only *approximately*, as long as sufficient progress is made at each stage. This makes our result more robust and practical. Formal statements and proofs are provided in §D. We also give non-asymptotic convergence rate in Theorem 2.

**Relation to Previous Work.** Prior iterative approaches to general-preference policy optimization—such as mirror descent-style algorithms (Azar et al., 2024; Wu et al., 2024)—typically only guarantee that the *average* iterate converges (in mixture) to a Nash policy. However, in practice, averaging across many deep neural network checkpoints is both storage- and deployment-inefficient and uncommon. Furthermore, existing methods for last-iterate convergence apply only to *regularized* games (Munos et al., 2024; Zhang et al., 2025b), yielding stationary points that may diverge from true Nash equilibria of the alignment game (see also Table 4). In contrast, COMAL is the first framework to attain fully practical and provable last-iterate convergence to the Nash equilibrium of the alignment game, even in large-scale LLM contexts. The convergence in alignment game without regularization is crucial to ensure 50% win rate against any other policy.

**Practical Instantiation.** Each COMAL iteration involves solving a regularized zero-sum game $J_\tau(\pi_1, \pi_2, \pi_{\mathrm{ref}})$, for which many policy optimization algorithms originally developed for RLHF and preference learning (e.g., PPO, DPO, IPO, INPO) can serve as efficient sub-solvers; see §3.2 for discussion and §F for variants. While the theoretical properties of COMAL provide a strong foundation, its practical implementation and empirical validation in large-scale LLM alignment constitute a central contribution of this work. We show that COMAL can be instantiated with, for example, *INPO* (Zhang et al., 2025b) as the regularized game solver (Algorithm 3), yielding substantial and consistent performance gains across challenging alignment benchmarks. Our results demonstrate that COMAL not only offers strong convergence guarantees but is also easy to deploy, highly scalable, and effective for real-world preference optimization and LLM fine-tuning. Notably, integrating COMAL into existing pipelines typically requires only minimal modifications—chiefly, adding periodic reference policy updates and an outer iteration loop—making it directly compatible with current large-scale alignment workflows. **We note that the per-iteration computational cost of our algorithm is comparable to other alignment algorithms tested in our experiments**—differing by only a few percent—while achieving better performance without significant computational overhead.

## 3.2 SOLVING A REGULARIZED GAME

Each iteration of COMAL requires solving a regularized zero-sum game $J_\tau(\pi_1, \pi_2, \pi_{\mathrm{ref}})$. We present Mirror Descent (MD) in Algorithm 2 for computing a Nash equilibrium of this game. MD builds on the prox operator, a principled tool from convex optimization that ensures stability and supports broad applicability. Importantly, we later show that this prox operator can be instantiated using a variety of modern policy optimization algorithms. For simplicity, we consider policies $\pi \in \Delta(\mathcal{Y})$ and omit dependence on the instruction $x$; all discussions extend naturally to the contextual setting.

**Mirror Descent and Multiplicative Weights Update (MWU).** Mirror Descent (MD) is a foundational family of iterative optimization algorithms, widely used in game theory, machine learning, and online learning (Nemirovskij & Yudin, 1983). At a high level, MD generalizes vanilla gradient descent by using a geometry-aware update rule that better respects the structure of the optimization domain through a more flexible notion of 'distance,' defined by a regularizer. A particularly important special case is the *Multiplicative Weights Update* (MWU) algorithm (Arora et al., 2012), which can be viewed as Mirror Descent performed with the negative entropy regularizer. For concreteness, suppose we want to maximize some smooth objective $f(\pi)$ over probabilistic policies $\pi$. At iteration

---

**Algorithm 2:** Regularized game solver for $J_\tau(\pi_1, \pi_2, \pi_{\text{ref}}) - \text{argmax}_{\pi_1} \min_{\pi_2} J_\tau(\pi_1, \pi_2, \pi_{\text{ref}})$

---

**Input:** Reference policy $\pi_{\text{ref}}$, preference oracle $\mathbb{P}$, regularization $\tau > 0$, step size $\eta > 0$, number
      of iterations $K \geq 1$
**Output:** Regularized Nash equilibrium policy $\mu_K$
Initialize $\mu^1 \leftarrow \pi_{\text{ref}}$
**for** $k = 1, 2, \ldots, K - 1$ **do**
    $\left|\right.$ $g_\tau^k \leftarrow \nabla_{\mu^k} J_\tau(\mu^k, \mu^k, \pi_{\text{ref}}) = \mathbb{P}(\cdot \succ \mu_k) - \tau(\log \frac{\mu_k(\cdot)}{\pi_{\text{ref}}(\cdot)} + 1)$ // Gradient
    $\left|\right.$ $\mu^{k+1} \leftarrow \text{Prox}(\mu_k, \eta g_\tau^k)$
**return** $\mu_K$

---

$t$, with current policy $\pi^t$, MWU computes the updated policy $\pi^{t+1}$ as the solution to:

$$\pi^{t+1} := \text{Prox}(\pi^t, \nabla f(\pi^T)) := \underset{\pi}{\text{argmax}} \left\{ \langle \nabla f(\pi^t), \pi \rangle - \eta^{-1} \cdot \text{KL}(\pi \| \pi^t) \right\}, \quad (3)$$

where $\eta$ is a positive parameter (step size), and $\text{KL}(\cdot \| \cdot)$ is the Kullback-Leibler (KL) divergence, which in this case measures how much the new policy deviates from the previous one. Intuitively, this update chooses a new policy by trading off following the gradient of $f$ with staying close (in KL) to the prior policy, preventing overly aggressive changes that could destabilize learning. This update can be viewed more generally through the lens of the *proximal operator* (or *prox operator*)—a mathematical abstraction that unifies many optimization steps used in machine learning, including projected gradient descent and mirror descent with Bregman divergences (Parikh et al., 2014). We include a detailed discussion on the prox operator in §C.

**Non-asymptotic Convergence.** Denote $\pi_\tau^\star$ the Nash equilibrium of the KL regularized game $J_\tau(\pi_1, \pi_2, \pi_{\text{ref}})$, which is $\tau$-strongly monotone. We can apply existing results to show that MWU (Algorithm 2) achieves linear last-iterate convergence rate: the KL divergence to the Nash equilibrium $\pi_\tau^\star$ decreases exponentially fast. The proof is in §E. Theorem 2 also implies **a non-asymptotic convergence to an approximate Nash equilibrium**: we can choose $\tau = O(\varepsilon)$ and approaching an $\varepsilon$-approximate Nash equilibrium of the original alignment game (1) in $\tilde{O}(1/\varepsilon^2)$ iterations.

**Theorem 2.** *For step size* $0 < \eta \leq \frac{\tau}{\tau^2 + 0.5}$, *Algorithm 2 guarantees for every* $k \geq 1$, $\text{KL}(\pi_\tau^\star \| \mu^{k+1}) \leq (1 - \frac{\eta\tau}{2})^k \text{KL}(\pi_\tau^\star \| \pi_{\text{ref}})$.

### 3.3 PRACTICAL METHODS FOR COMPUTING THE PROX OPERATOR

We show how to implement COMAL in practical large-scale applications like LLM alignment by computing the prox operator, with a concrete implementation presented in **Algorithm 3** . Specifically, we observe that many existing algorithms designed for RLHF and preference optimization with neural network parameters can be extended to solve the prox operator . These algorithms include RL algorithms like PPO (Schulman et al., 2017) and GRPO (Shao et al., 2024; Guo et al., 2025) and loss-minimization algorithms like, DPO (Rafailov et al., 2024), IPO (Azar et al., 2024), SPPO (Wu et al., 2024), REBEL (Gao et al., 2024), DRO (Richemond et al., 2024), INPO (Zhang et al., 2025b). Each of them may be preferred in certain settings. Due to space limitations, we defer the detailed discussion to §F. We also note that our meta algorithm, COMAL, can be integrated with many existing methods designed for preference optimization with minimal change, and we present concrete instantiations of COMAL using iterative GRPO, SPPO, REBEL, and DRO in §G.

Our unified view on existing diverse preference methods through the perspective of computing the prox operator opens the possibility of applying other algorithms from online learning and optimization to robust LLM alignment. We include implementations for two other last-iterate convergent algorithms, the Mirror-Prox algorithm (Nemirovski, 2004) and the Optimistic Multiplicative Weights Update algorithm (Rakhlin & Sridharan, 2013; Syrgkanis et al., 2015), in §H.

## 4 SYNTHETIC EXPERIMENTS

We conduct experiments on a simple bandit problem with $\mathcal{Y} = \{y_a, y_b, y_c\}$ and non-BT preference model over $\mathcal{Y}$. Specifically, we set $\mathbb{P}[y_b \succ y_a] = \mathbb{P}[y_c \succ y_b] = 0.9$ and $\mathbb{P}[y_a \succ y_c] = 0.8$. Observe

---

**Algorithm 3:** Practical Implementation of COMAL integrated with INPO (Algorithm 4)

---

**Input:** Initial policy $\pi_{\text{init}}$, regularization $\{\tau_t > 0\}$, step size $\{\eta_t > 0\}$, number of outer
       iterations $T \geq 1$, number of inner iterations $\{K_t \geq 1\}$, preference oracle $\mathbb{P}$.
**Output:** Optimized policy $\pi^T$
Initialize $\pi^1, \pi_{\text{ref}} \leftarrow \pi_{\text{init}}$
**for** $t = 1, 2, \ldots, T - 1$ **do**
    |  $\pi^{t+1} \leftarrow \text{INPO}(\pi_{\text{ref}}, \tau_t, \eta_t, K_t, \mathbb{P})$ (Algorithm 4)
    |  $\pi_{\text{ref}} \leftarrow \pi^{t+1}$
**return** $\pi^T$

---

that the preference is intransitive and exhibits a preference cycle $y_c \succ y_b \succ y_a \succ y_c$. The detailed setup and result analysis are in §I and Figure 1, 3, and 4. Due to the space limit, we only briefly discuss the results here. Our experiments show that iterative DPO, iterative IPO (Azar et al., 2024), and SPPO (Wu et al., 2024) all cycle and diverge away from the unique Nash equilibrium. The INPO algorithm converges in the modified game as we show in Theorem 2. However, the converging point is not the Nash equilibrium of the original game and suffers a constant equilibrium gap. COMAL is the only algorithm that converges to the Nash equilibrium.

## 5 LLM-BASED EXPERIMENTS

We conduct experiments based on Llama-3-8B-Instruct (Dubey et al., 2024) and Qwen2.5-7B (Yang et al., 2024b),[3] on a commonly used dataset UltraFeedback (Cui et al., 2023) to show the effectiveness of COMAL under the practical preference optimization setting, following Algorithm 3.

### 5.1 EXPERIMENTAL SETTINGS

**Instruction Set.** Our training experiments are conducted on the 64K instructions from the UltraFeedback dataset, which covers a broad range of instruction types and is well-suited and widely used for studying preference optimization in practical scenarios.

**Preference Oracle.** We choose a mixture of two BT reward models as the preference oracle to simulate the preference diversity among human annotators. Specifically, the win rate of an output $y_a$ over $y_b$ parameterized by a mixture of two BT reward models $r_1$ and $r_2$ is

$$P(y_a > y_b) = \frac{1}{2} \cdot \frac{e^{r_1(y_a)}}{e^{r_1(y_a)} + e^{r_1(y_b)}} + \frac{1}{2} \cdot \frac{e^{r_2(y_a)}}{e^{r_2(y_a)} + e^{r_2(y_b)}}. \tag{4}$$

The two reward models used are Skywork-Reward-Llama-3.1-8B-v0.2 (Liu et al., 2024) and ArmoRM-Llama3-8B-v0.1 (Wang et al., 2024), both achieving strong performance on various human preference alignment benchmarks in RewardBench (Lambert et al., 2024b).

**Preference Data Generation.** To construct the preference data, i.e., output pairs with a preference annotation specifying which one is better, we adopt the setting of Zhang et al. (2025b) by sampling 5 candidate outputs for each instruction with a temperature of 0.8 and applying the preference oracle to select the best and the worst candidates to form a data point.

**Baselines.** The following baselines are compared: (1) **BASE**: Llama-3-8B-Instruct, which has already been fine-tuned, can be directly used as the base model following SimPO (Meng et al., 2024). For Qwen2.5-7B, we finetune it using the standard SFT objective on the Tulu3 SFT dataset (Lambert et al., 2024a). (2) vanilla **DPO** (Rafailov et al., 2024) and (3) vanilla **IPO** (Azar et al., 2024), where one training iteration is performed over the entire instruction set of UltraFeedback with output pairs sampled from the BASE policy; (5) **INPO** (Zhang et al., 2025b)(Algorithm 4), where each iteration of training is performed on a single data subset; (6) **Iterative IPO (Iter-IPO)**, which follows a training setting similar to INPO but without the KL regularization with respect to the static reference policy.

**Training Details.** To reduce computational cost, the instructions in UltraFeedBack are divided into six equal subsets (10K each), with one subset used per training iteration. For iterative optimization

---

[3]Additional experiments based on Qwen2-1.5B (Yang et al., 2024a) are also provided in the §K.

algorithms, 18 training iterations are performed. All iterative optimization algorithms compared have similar computational costs, each taking around 100 hours on 8 NVIDIA A6000 GPUs. To the best of our knowledge, multi-iteration training like ours has rarely been explored in previous work. For example, INPO only trained up to 3 iterations, equivalent to just one full round over UltraFeedback's instructions. The overall update process is as follows: (1) **Iter-IPO**: at each iteration, the reference policy in the IPO loss (Eq. 13) is updated to the policy produced in the previous iteration; (2) **INPO**: at each iteration, one optimization step in Algorithm 4 is performed, with the reference policy fixed to the BASE policy; (3) **COMAL**: as outlined in Algorithm 3, COMAL uses INPO as a sub-routine, and updates the reference policy in INPO every 6 iterations, i.e., an entire pass of the instruction set.

**Hyper-Parameters.** We conduct a grid search for the KL regularization strength, $\eta^{-1}$, for DPO, IPO and INPO, within the range of 0.001 - 0.1. The value of $\tau$ in INPO (Equation (14)) is determined by following Zhang et al. (2025b), where $\eta\tau$ is set to a fixed ratio, $1/3$. We found **Iter-IPO** and **INPO** achieve the best performance when $\eta^{-1}$ is 0.002. However, in Llama-3 training, we observe rapid performance degradation of both algorithms after 6 training iterations. In contrast, training Qwen2.5-SFT remains stable. We posit that this is because Llama-3-8B-Instruct has undergone more extensive post-training, making further updates more intricate. We then explored larger values of $\eta^{-1}$ for stable training and found that it must be increased to around 0.1 to maintain stability after 6 iterations. Therefore, to study the algorithms' behavior in more training iterations, we perform additional experiments with $\eta^{-1}$ set to 0.1 (**Iter-IPO-L** and **INPO-L**), which leads to stabler training. For **COMAL**, since it involves multi-round INPO training with adjustable KL regularization strengths (Algorithm 3), we set $\eta^{-1}$ to 0.002 for the first INPO training round (i.e., 6 iterations) and adjust it to 0.1 for the subsequent two rounds, balancing training stability with efficiency. In Qwen2.5 training, $\eta^{-1}$ is fixed to 0.002 for all algorithms since the training process remains stable. More details are in Appendix J.

**Evaluations.** We use the instructions in a widely used benchmark, AlpacaEval (Li et al., 2023), to construct the test set, since these instructions cover various task scenarios. However, instead of using GPT-4, the default evaluator for the AlpacaEval benchmark, **we chose to use the same preference oracle used during training as the evaluator**. This follows the setting of previous work (Munos et al., 2024), which provides a controlled experimental setting, ensuring that the preference oracle the model learns to fit is also the one used to evaluate its performance.

## 5.2 RESULT ANALYSIS

Table 1: Performance comparison of different training algorithms evaluated by the preference oracle. The row v.s. column win rate (%) is reported. All the training is based on the **BASE** checkpoint, `Llama-3-8B-Instruct`. For Iterative IPO (**Iter-IPO**) and **INPO**, we report their performance with both a small, optimal regularization ($\eta^{-1} = 0.002$) after 6 iterations and a large regularization ($\eta^{-1} = 0.1$, **Iter-IPO-L** and **INPO-L**) after 18 iterations.

| Row/Column | BASE | IPO | DPO | Iter-IPO-L | Iter-IPO | INPO-L | INPO | COMAL | Avg |
|---|---|---|---|---|---|---|---|---|---|
| IPO | 93.04 | 50.00 | 47.20 | 28.20 | 20.75 | 83.23 | 25.22 | 21.61 | 46.16 |
| DPO | 92.42 | 52.80 | 50.00 | 28.57 | 21.37 | 81.49 | 26.46 | 21.37 | 46.81 |
| Iter-IPO | **94.16** | **79.25** | **78.63** | 50.68 | 50.00 | **89.19** | 53.79 | 39.75 | 66.93 |
| INPO | 92.92 | 74.78 | 73.54 | 47.08 | 46.21 | 87.20 | 50.00 | 35.78 | 63.44 |
| COMAL | 90.43 | 78.39 | **78.63** | **62.98** | **60.25** | 86.09 | **64.22** | **50.00** | **71.37** |

Table 1 and Table 2 perform pairwise comparisons of different algorithms. For **Iter-IPO** and **INPO**, we evaluate their *best* checkpoints due to significant performance degradation thereafter. For **Iter-IPO-L**, **INPO-L**, and **COMAL**, comparisons are made at the final 18-iteration checkpoint. The result shows that **COMAL achieves a win rate exceeding 60.2% against all competing algorithms when using Llama-3-8B-Instruct, and 56.9% with Qwen2.5-7B,** demonstrating its effectiveness.

Figure 2 presents the training dynamics of three iterative preference optimization algorithms, where the average win rate is computed against all the algorithms in Table 1 and Table 2. We note that:

Table 2: Performance comparison of different training algorithms evaluated by the preference oracle. The row v.s. column win rate (%) is reported. All the training is based on the **BASE** checkpoint, which is fine-tuned from `Qwen2.5-7B` using the SFT objective.

| Row/Column | BASE | IPO | DPO | Iter-IPO | INPO | COMAL | Avg |
|---|---|---|---|---|---|---|---|
| IPO | 91.43 | 50.00 | 50.19 | 22.98 | 23.73 | 21.37 | 43.28 |
| DPO | 90.68 | 49.81 | 50.00 | 23.35 | 23.60 | 20.50 | 42.99 |
| Iter-IPO | **91.68** | 77.02 | 76.65 | 50.00 | 50.43 | 43.11 | 64.81 |
| INPO | 90.81 | 76.27 | 76.40 | 49.57 | 50.00 | 42.11 | 64.19 |
| COMAL | 90.68 | **78.63** | **79.50** | **56.89** | **57.89** | 50.00 | **68.93** |

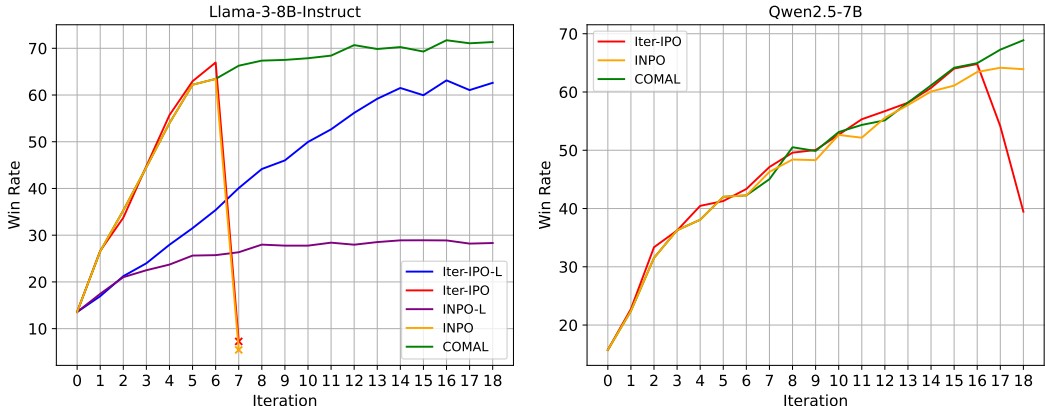

Figure 2: Comparisons of Iterative IPO (Iter-IPO), INPO, and COMAL. The average win rates of the trained checkpoints at each iteration against each training algorithm are displayed.

(1) COMAL consistently outperforms other algorithms, **showing steady improvements even in the late stages of the training period.**

(2) Both Iter-IPO and INPO exhibit rapid degradation at the 7th training iteration in Llama-3 training. We posit that this is because Llama-3-8B-Instruct has already undergone extensive post-training, making further optimization more delicate. Training with a larger KL-regularization with Llama-3-8B-Instruct leads to stabler training for both Iter-IPO(-L) and INPO(-L). However, it also introduces a lower performance upper bound. As discussed above, COMAL overcomes this limitation by dynamically adjusting the strength of the KL-regularization.

**Evaluation Results on Standard Benchmarks.** To verify that the checkpoints produced by our algorithm retain general capabilities, we compare their performance against the baselines on six standard LLM benchmarks as a sanity check. These include GSM8K for math problem solving (Cobbe et al., 2021), MMLU for multi-task language understanding (Hendrycks et al., 2021), BigBench Hard (BBH) for reasoning (Suzgun et al., 2023), HumanEval for coding (Chen et al., 2021), and two LLM alignment evaluation benchmarks, AlpacaEval and Arena-Hard, where the original evaluator, GPT-4, is used. The results in Table 3, highlighting two findings:

(1) **COMAL maintains comparable performance on standard academic benchmarks**; (2) While not optimized for GPT-4's preferences, **COMAL performs strongly on AlpacaEval and Arena-Hard compared to the baselines**, indicating its generalizability. We note that COMAL does not outperform Iter-IPO on Arena-Hard. However, as noted above, we compare Iter-IPO at its best checkpoint, whereas COMAL is evaluated at the final checkpoint, because Iter-IPO's performance declines near the end of training (Figure 2). Moreover, since Arena-Hard compares each model only against a fixed baseline (GPT-4), its setup does not fully align with COMAL's objective.

**Discussion on Updating the Reference Policy.** Our theoretical analysis in Section 3 indicates the reference policy in COMAL's objective needs to be updated in order to converge to the alignment

Table 3: Performance of various preference optimization algorithms on standard benchmarks.

| Method | GSM8K | MMLU | BBH | HumanEval | AlpacaEval | Arena-Hard |
|---|---|---|---|---|---|---|
| BASE (Llama-3-8B-Instruct) | 77.0 | 63.5 | 66.4 | 79.2 | 25.0 | 21.3 |
| IPO(-Llama-3-8B-Instruct) | 74.0 | 63.1 | 53.4 | 65.3 | 48.7 | 38.9 |
| DPO(-Llama-3-8B-Instruct) | 73.5 | 62.7 | 51.0 | 65.6 | 48.6 | 33.0 |
| Iter-IPO(-Llama-3-8B-Instruct) | 71.5 | 64.4 | 62.6 | 77.7 | 50.6 | 43.8 |
| INPO(-Llama-3-8B-Instruct) | 73.0 | 64.7 | 61.7 | 77.4 | 51.6 | 41.0 |
| COMAL(-Llama-3-8B-Instruct) | 77.5 | 64.9 | 63.3 | 77.2 | 53.5 | 41.3 |
| BASE (Qwen2.5-7B-SFT) | 77.5 | 70.9 | 65.6 | 84.0 | 14.7 | 22.3 |
| IPO(-Qwen2.5-7B-SFT) | 91.0 | 70.2 | 67.2 | 86.7 | 33.4 | 53.2 |
| DPO(-Qwen2.5-7B-SFT) | 91.0 | 70.2 | 66.9 | 86.6 | 34.8 | 54.3 |
| Iter-IPO(-Qwen2.5-7B-SFT) | 91.0 | 70.8 | 71.9 | 86.3 | 42.9 | 64.5 |
| INPO(-Qwen2.5-7B-SFT) | 91.5 | 70.7 | 71.0 | 86.8 | 39.8 | 62.2 |
| COMAL(-Qwen2.5-7B-SFT) | 91.0 | 70.8 | 72.3 | 85.1 | 42.2 | 63.0 |

game (Equation (1)). Emprically, it means that COMAL does not have a KL-regularization from a static reference policy. However, as shown in Table 3, COMAL does not suffer substantially from the "alignment tax" (Dong et al., 2024; Ouyang et al., 2022). Moreover, we observe that its improvement is not solely from relaxing the KL-constraint – Iter-IPO has even smaller constraints from a reference policy updated at each iteration, but fails to outperform COMAL and suffers from training instability.

## 6 CONCLUSION

We have proposed COMAL, a meta-algorithm for preference optimization that provably converges to the Nash equilibrium policy in the last iterate. We have provided a theoretical analysis of the properties of COMAL and have empirically demonstrated its effectiveness under both synthetic and real-world experimental settings, where COMAL consistently maintains a win rate above 50% against other policies in controlled settings. We believe COMAL has significant potential to enhance the performance of LLMs in the alignment fine-tuning setting, due to its theoretical guarantees and flexibility, as it can be integrated with existing learning algorithms while overcoming their limitations.

## ACKNOWLEDGEMENTS

We are grateful for the TPU compute support provided by the Google TRC program and for the OpenAI API credits support provided by OpenAI's Researcher Access Program. YC is supported by the NSF Award CCF-2342642. AO is supported by a Meta Ph.D. Fellowship. WZ is supported by the NSF Award CCF-2342642 and a Research Fellowship from the Center for Algorithms, Data, and Market Design at Yale (CADMY).

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

## CONTENTS

## A  RELATED WORK

**Alignment under Preference models**  Most existing approaches adopt the Bradley-Terry (BT) preference model (Bradley & Terry, 1952; Christiano et al., 2017), which involves first learning a preference model and then optimizing the objective function with a KL divergence penalty relative to the original language model. For example, RLHF (Ouyang et al., 2022) aims to ensure that LLMs follow instructions by initially learning a BT model and subsequently fine-tuning the model based on the learned reward while regularizing it with the original LLM.

Building on this framework, Rafailov et al. (2024) introduces Direct Preference Optimization (DPO) that maintains the assumption of the BT model for preferences but eliminates the preference learning step by reformulating the objective and optimizing it directly. Additionally, Ethayarajh et al. (2024) diverges from the traditional BT-based methods by deriving algorithms that bypass the preference modeling step altogether. Instead, they model user preferences based on Kahneman and Tversky's utility theory.

**Alignment Solution Concepts under General Preferences**    Azar et al. (2024) is the first to consider general preferences. They propose the IPO algorithm, an offline algorithm that directly optimizes the win rate of the model penalized by the KL divergence with respect to the original model. Munos et al. (2024) also consider general preferences and aim to find the *von Neumann winner*, which corresponds to the Nash equilibrium of a game played between the two LLMs over the win rate. They propose a variant of the Mirror Descent (MD) algorithm called Nash-MD and show last-iterate convergence in the KL-regularized game. Concurrently, Swamy et al. (2024) study the same solution concept focusing more on sequential games. Calandriello et al. (2024) proved that the objective of the the IPO algorithm coincides with the Nash policy under a proper choice of the parameter that controls the regularization. The work of Liu et al. (2025a) further studies the statistical properties of the Nash equilibrium policy, showing that Nash equilibria correspond to Condorcet winners if they exist, and if not, the Nash equilibrium must be mixed. These results show the importance of finding the Nash equilibrium of the original game rather than the KL-regularized game. The work by Pásztor et al. (2025) proposes Stackelberg learning from human feedback, aiming to find the Stackelberg equilibrium of a two-player sequential-move game.

**Iterative Self-Play Algorithms**    Apart from the aforementioned works, recent research has also proposed practical implementations of Mirror Descent (MD) algorithms, which can be used to learn Nash equilibria through self-play. Rosset et al. (2024) propose Direct Nash Optimization (DNO), where, at each iteration, the model regresses predicted preferences against actual preferences using cross-entropy loss. Similarly, Wu et al. (2024) introduces the Self-Play Preference Optimization (SPPO) method, Gao et al. (2024) introduces Reinforcement Learning via Regressing Relative Rewards (REBEL), and Richemond et al. (2024) introduces the Direct Reward Optimization (DRO) which regresses the loss using the $\ell_2$ distance at each iteration. Since these algorithms simulate the MD update, when applied in a two-player zero-sum game, they only have average-iterate convergence but all *diverge in the last iterate*. Moreover, all these methods require the estimation of the win rate, which can be computationally expensive.

Most closely related to our work is Iterative Nash Policy Optimization (INPO) by Zhang et al. (2025b), which continues to use $\ell_2$ distance regression. However, by further reformulating and simplifying the objective in a manner similar to IPO, INPO eliminates the need to estimate the expected win rate. The primary distinction between our approach and INPO is that INPO is designed for the KL-regularized game and is equivalent to MD; while our algorithm COMAL is inspired by the Conceptual Prox algorithm and guarantees last-iterate convergence in the original game. This fundamental difference allows COMAL to achieve more favorable convergence properties and outperform INPO, achieving a win rate strictly greater than 50% against it. The work by Tang et al. (2025) also proposes methods for the regularized game, which do not achieve last-iterate convergence in the original game.

**Last-Iterate Convergence in Games**    Mirror Descent fails to converge in simple zero-sum games, often resulting in cycling behavior (Mertikopoulos et al., 2018). In contrast, several algorithms have been shown to achieve last-iterate convergence including the Proximal Point (PP) method (Rockafellar, 1976), Extra-Gradient (EG) (Korpelevich, 1976), Optimistic Gradient Descent (OGD) (Popov, 1980; Rakhlin & Sridharan, 2013), and the Conceptual Prox/Mirror Prox methods (Nemirovski, 2004). The asymptotic convergence properties of these algorithms have been extensively studied (Popov, 1980; Facchinei & Pang, 2003; Iusem et al., 2003; Nemirovski, 2004; Daskalakis & Panageas, 2018). Recently, there has been a growing focus on establishing finite-time convergence guarantees for these methods, addressing the practical necessity of understanding their performance within a limited number of iterations (see e.g., (Mokhtari et al., 2020b;a; Golowich et al., 2020b;a; Perolat et al., 2021; Bauschke et al., 2021; Wei et al., 2021; Cai et al., 2022; Gorbunov et al., 2022; Cai & Zheng, 2023a;b; Cai et al., 2023; Abe et al., 2024; Cai et al., 2024b;a) and references therein). In particular, Perolat et al. (2021); Abe et al. (2024); Sokota et al. (2023) propose algorithms that are variants of the Conceptual-Prox algorithm (Nemirovski, 2004) and achieve last-iterate convergence under the

Table 4: Property comparison of different preference optimization algorithms. The algorithms are compared based on whether they work for **general preferences** and whether they exhibit **last-iterate convergence** in two-player zero-sum games. ✗ : convergence only in the modified KL-regularized game $J_\tau(\pi_1, \pi_2, \pi_{\text{ref}})$ (2) but not in $J(\pi_1, \pi_2)$ (1).

| Algorithm | General Preference | Convergence |
|---|---|---|
| DPO (Rafailov et al., 2024) IPO (Azar et al., 2024) | ✗ | ✗ |
| SPPO (Wu et al., 2024) Nash-MD (Munos et al., 2024) | ✓ | ✗ |
| INPO (Zhang et al., 2025b) | ✓ | ✗ |
| COMAL (Algorithm 1) | ✓ | ✓ |

assumption the regularized game can be solved exactly. Our work further extends their results to the case where the regularized game can be solved only approximately and demonstrates COMAL's effectiveness in large-scale LLM alignment setting.

While our work focuses on the Conceptual-Prox algorithm, in §H we also include practical implementations of other convergent methods, including the mirror-prox method (Nemirovski, 2004) that generalizes the extragradient method (Korpelevich, 1976), and the Optimistic Multiplicative Weight Update algorithm (Rakhlin & Sridharan, 2013). We remark that several concurrent and subsequent works (Zhou et al., 2025; Zhang et al., 2025a; Wu et al., 2025; Tiapkin et al., 2025) have also investigated both the theoretical and practical performance of Mirror-Prox (which subsumes the extragradient method) and OMWU for LLM alignment. Taken together with our experiments, these studies provide extensive evidence that provably last-iterate convergent algorithms are effective for LLM alignment.

**Comparison with (Wang et al., 2025)**   The concurrent and independent work by Wang et al. (2025) also presents a last-iterate convergent method for NLHF. Their algorithm is based on the Magnetic Mirror Descent (MMD) method (Sokota et al., 2023), which is also an implementation of the conceptual prox algorithm. The main differences between the two works are:

- The theoretical results of last-iterate convergence in (Wang et al., 2025) require solving each regularized game *exactly*. This requires an infinite number of iterations to solve each subgame (see their Theorems 3.4 and 3.7). In contrast, we prove last-iterate convergence under the weaker assumption that each regularized game is solved *approximately*.

- The experiments in (Wang et al., 2025) compare only reward-based methods such as PPO and DPO, while we conduct extensive experiments comparing COMAL with both DPO and methods for general preferences and NLHF, such as iterative IPO and INPO. While Wang et al. (2025) only report the win rate of their method against the base model, we report improved win rates against the base model and across all baseline methods, including DPO, IPO, and INPO. Our results show that COMAL achieves a consistent > 50% win rate against all baseline models, which is an important property of Nash equilibrium convergence.

- Wang et al. (2025) does not report results on standard benchmarks such as Arena-Hard or Alpaca-Eval 2, we present a comprehensive evaluation of COMAL and baseline methods on these benchmarks. Our results show the robustness of COMAL and that the alignment tax is very mild.

## B   ADDITIONAL BACKGROUNDS

A special case of the general preference model is the Bradley-Terry (BT) model, which assumes a reward function parameterizes the preference.

**Definition 3** (Bradley-Terry Model). *A preference model $\mathbb{P}$ satisfies the* Bradley-Terry (BT) *assumption if there exists a reward function $r^* : \mathcal{X} \times \mathcal{Y} \to \mathbb{R}$ such that*

$$\mathbb{P}(y_1 \succ y_2 \mid x) = \frac{\exp\left(r^*(x, y_1)\right)}{\exp\left(r^*(x, y_1)\right) + \exp\left(r^*(x, y_2)\right)}$$
$$= \sigma(r^*(x, y_1) - r^*(x, y_2)).$$

### B.1 ALIGNMENT UNDER THE BRADLEY-TERRY MODEL ASSUMPTION

**RLHF** Reinforcement Learning from Human Feedback (RLHF) is to first learn a reward function $r$ under the BT model and then find the optimal KL regularized policy $\pi^*$ w.r.t. the learned reward function $r$:

$$\pi^* := \arg\max_{\pi} \mathbb{E}_{x\sim\rho, y\sim\pi(\cdot|x)}$$
$$\left[ r(x, y) - \eta^{-1} \mathrm{KL}(\pi(\cdot \mid x) || \pi_{\mathrm{ref}}(\cdot \mid x)) \right], \tag{5}$$

where $\eta^{-1} > 0$ controls the regularization, and $\pi_{\mathrm{ref}}$ is the initial reference model, usually the policy $\pi_{\mathrm{sft}}$ obtained from pre-training and supervised fine-tuning.

**DPO** Rafailov et al. (2024) observe that the regularized optimization problem (5) has a closed-form solution: for any $x$ and $y$, $\pi^*(y \mid x) = \frac{\pi_{\mathrm{ref}}(y|x)\exp(\eta r(x,y))}{Z_x}$, where $Z_x = \mathbb{E}_{y\sim\pi_{\mathrm{ref}}(\cdot|x)}[\exp(\frac{1}{\eta}r(y,x))]$ is the normalization constant known as the partition function. Since $\pi^*$ implicitly parameterizes the reward function $r$. Rafailov et al. (2024) propose direct preference optimization (DPO) to learn the optimal policy using the maximum likelihood objective directly:

$$\ell_{\mathrm{DPO}}(\pi; \pi_{\mathrm{ref}}) = -\mathbb{E}_{(x,y_w,y_l)\sim\mathcal{D}}$$
$$\left[ \log \sigma \left( \eta^{-1} \log \frac{\pi(y_w \mid x)}{\pi_{\mathrm{ref}}(y_w \mid x)} - \eta^{-1} \log \frac{\pi(y_l \mid x)}{\pi_{\mathrm{ref}}(y_l \mid x)} \right) \right],$$

where $\mathcal{D}$ is a data set containing win-loss pair of responses $\{y_w, y_l\}$ given prompt $x$.

### B.2 CAST STUDY: A SINGLE BT MODEL CANNOT REPRESENT A MIXTURE OF TWO BT MODELS (ADDED DURING REBUTTAL PERIOD)

We would like to note that a mixture of two BT reward models can represent complex preference patterns that exhibit intransitivity or even cyclic preferences, and in general does not satisfy the BT assumption in Definition 3. Consider a fixed prompt $x \in \mathcal{X}$ and three candidate responses $A, B, C \in \mathcal{Y}$ (for brevity, we omit $x$ from the notation and write $r^*(A)$ instead of $r^*(x, A)$). Define two BT preference models $\mathbb{P}_1, \mathbb{P}_2$ with corresponding BT reward functions $r_1^*, r_2^*$ satisfying

$$r_1^*(A) = 1, \quad r_1^*(B) = 2, \quad r_1^*(C) = 3,$$

$$r_2^*(A) = 5, \quad r_2^*(B) = 3, \quad r_2^*(C) = 1.$$

By the BT assumption,

$$\mathbb{P}_k(i \succ j \mid x) = \sigma(r_k^*(i) - r_k^*(j)), \qquad k \in \{1, 2\}, \ i, j \in \{A, B, C\},$$

where $\sigma(t) = \frac{1}{1+e^{-t}}$.

**Cyclic preferences from a mixture.** Consider first the mixture preference model

$$\mathbb{P}_{\mathrm{mix}}(i \succ j \mid x) := 0.6 \, \mathbb{P}_1(i \succ j \mid x) + 0.4 \, \mathbb{P}_2(i \succ j \mid x).$$

A direct calculation yields

$$\mathbb{P}_{\mathrm{mix}}(A \succ B \mid x) \approx 0.514 > 0.5, \quad \mathbb{P}_{\mathrm{mix}}(B \succ C \mid x) \approx 0.514 > 0.5,$$

$$\mathbb{P}_{\mathrm{mix}}(C \succ A \mid x) \approx 0.536 > 0.5.$$

Thus the induced preference is cyclic:

$$A \succ B, \quad B \succ C, \quad C \succ A.$$

In particular, the mixture $\mathbb{P}_{\mathrm{mix}}$ violates transitivity, even though each component $\mathbb{P}_1, \mathbb{P}_2$ individually satisfies the BT assumption.

**Equal-weight mixture is not BT-representable.** Now consider the equal-weight mixture

$$\mathbb{P}_{\mathrm{mix}}(i \succ j \mid x) := 0.5\,\mathbb{P}_1(i \succ j \mid x) + 0.5\,\mathbb{P}_2(i \succ j \mid x).$$

In this case we obtain

$$\mathbb{P}_{\mathrm{mix}}(A \succ B \mid x) \approx \mathbb{P}_{\mathrm{mix}}(B \succ C \mid x) \approx 0.575 > 0.5, \qquad \mathbb{P}_{\mathrm{mix}}(A \succ C \mid x) \approx 0.551.$$

We now show that these three probabilities cannot arise from any single BT model. Suppose, for contradiction, that there exists a reward function $\tilde{r}^* : \mathcal{X} \times \mathcal{Y} \to \mathbb{R}$ such that the corresponding BT model $\tilde{\mathbb{P}}$ satisfies

$$\tilde{\mathbb{P}}(i \succ j \mid x) = \sigma(\tilde{r}^*(i) - \tilde{r}^*(j)),$$

and matches the mixture probabilities for $i, j \in \{A, B, C\}$. Since

$$\tilde{\mathbb{P}}(A \succ B \mid x) \approx \tilde{\mathbb{P}}(B \succ C \mid x) \approx 0.575 > 0.5,$$

and $\sigma$ is strictly increasing with $\sigma(0) = 0.5$, we must have

$$\tilde{r}^*(A) - \tilde{r}^*(B) = \tilde{r}^*(B) - \tilde{r}^*(C) =: d > 0.$$

Therefore

$$\tilde{r}^*(A) - \tilde{r}^*(C) = (\tilde{r}^*(A) - \tilde{r}^*(B)) + (\tilde{r}^*(B) - \tilde{r}^*(C)) = 2d > d,$$

which implies, by strict monotonicity of $\sigma$,

$$\tilde{\mathbb{P}}(A \succ C \mid x) = \sigma(\tilde{r}^*(A) - \tilde{r}^*(C)) = \sigma(2d) > \sigma(d) = \tilde{\mathbb{P}}(A \succ B \mid x) \approx 0.575.$$

However, the mixture model satisfies

$$\mathbb{P}_{\mathrm{mix}}(A \succ C \mid x) \approx 0.551 < 0.575,$$

a contradiction. Hence the 50%-50% mixture of these two BT models cannot be represented by any single BT model satisfying the BT assumption.

## C  PROX OPERATOR

**Prox Operator and Bregman Divergence.** To define the prox operator, we first introduce the *Bregman divergence*, which generalizes the notion of squared distance. For a convex function $\varphi : \mathcal{Z} \to \mathbb{R}$ (called the regularizer), the Bregman divergence between $z$ and $z'$ is defined as $D_\varphi(z\|z') := \varphi(z) - \varphi(z') - \langle \nabla\varphi(z'), z - z' \rangle$, where $\nabla\varphi(z')$ is the gradient of $\varphi$ at $z'$. The *prox operator* then takes a current point $z \in \mathcal{Z}$ and a (sub)gradient direction $g \in \mathbb{R}^n$, and returns the next point according to:

$$\mathrm{Prox}(z, g) := \underset{z'}{\mathrm{argmax}} \left\{ \langle g, z' \rangle - D_\varphi(z'\|z) \right\}.$$

This formulation interpolates between moving in the direction of $g$ and staying close to $z$, as measured by the Bregman divergence for the chosen regularizer. Two important *special cases* are

- When $\varphi(z) = \frac{1}{2}\|z\|^2$ (the squared Euclidean norm), $D_\varphi$ is just the squared distance, and the prox operator reduces to the usual projected gradient step.
- When $\varphi(z)$ is the negative entropy (as in MWU), the Bregman divergence is the KL divergence, leading to updates appropriate for probability distributions.

In our framework, we will instantiate the prox operator with choices of $\varphi$ and $g$ that map directly onto concrete policy-learning algorithms. In this paper, when we refer to the prox operator, we use the negative entropy regularizer $\varphi(z) = \sum_{i=1}^{n} z[i] \ln z[i]$, for which the corresponding Bregman divergence $D_\varphi$ is the KL divergence. Under this choice, the MWU update in Equation (3) is equivalent to the prox-form update $\pi^{t+1} = \mathrm{Prox}(\pi^t, \eta\nabla f(\pi^t))$.

### C.1  PROPERTIES OF THE PROX OPERATOR

Recall that $\mathrm{Prox}(z, g) = \mathrm{argmax}_{z' \in \mathcal{Z}} \langle g, z' \rangle - D_\varphi(z'\|z) = \mathrm{argmax}_{z' \in \mathcal{Z}} \langle g + \nabla\varphi(z), z' \rangle - \varphi(z')$. The following properties of the prox operator are well-known in the literature(e.g., (Nemirovski, 2004))

**Lemma 1.** $\mathrm{Prox}(z, g) = z'$ *if and only if* $\langle g + \nabla\varphi(z) - \nabla\varphi(z'), z' - z^* \rangle \geq 0$ *for all* $z^* \in \mathcal{Z}$.

**Corollary 1.** *Let* $\mathrm{Prox}(z, g) = z'$, *then*

$$\langle g, z^* - z' \rangle \leq D_\varphi(z^*\|z) - D_\varphi(z^*\|z') - D_\varphi(z'\|z), \quad \forall z^* \in \mathcal{Z}$$

# D    LAST-ITERATE CONVERGENCE OF COMAL

The proof of Theorem 1 is largely inspired by existing results for the conceptual prox algorithm in the literature (Facchinei & Pang, 2003; Nemirovski, 2004). We first consider the case where each step of COMAL, $\pi^{t+1} \leftarrow \operatorname{argmax}_{\pi_1} \min_{\pi_2} J_\tau(\pi_1, \pi_2, \pi_{\mathrm{ref}})$, can be solved *exactly* in Appendix D.1. We then extend the proof to the case where we only solve the regularized game *approximately* in Appendix D.2. In both cases, we prove last-iterate convergence to Nash equilibrium, i.e., $\lim_{t\to\infty} \pi^t$ exists and is a Nash equilibrium. The proof for the latter case seems to be the first in the literature.

In Theorem 1, we make the following assumption.

**Assumption 1.** *We assume there exists a Nash equilibrium $\pi^\star$ such that $\operatorname{supp}(\pi^\star) = \operatorname{supp}(\pi_{\mathrm{init}})$.*

This assumption is mild and **much weaker** than the "Bounded Log Density" assumptions used in previous works (Rosset et al., 2024; Zhang et al., 2025b), which directly assumes $|\log \frac{\pi^t}{\pi_{\mathrm{init}}}|$ is bounded.

## D.1    LAST-ITERATE CONVERGENCE UNDER EXACT SOLUTIONS

Recall that $\Pi := \{\pi : \operatorname{supp}(\pi) \subseteq \operatorname{supp}(\pi_{\mathrm{init}})\}$. Then $\mathrm{KL}(\pi||\pi_{\mathrm{init}}) \leq D := \max_{y:\pi_{\mathrm{init}}(y)>0} \log \pi_{\mathrm{init}}(y)$ is bounded for any $\pi \in \Pi$. We first prove $\mathrm{KL}(\pi^\star||\pi^{t+1}) \leq \mathrm{KL}(\pi^\star||\pi^t)$ for any $t \geq 1$.

**Lemma 2.** *Let $\pi^\star$ be an Nash equilibrium of $J(\pi_1, \pi_2)$. Then for any $\tau > 0$, if*

$$(\pi, \pi) = \operatorname*{argmax}_{\pi_1 \in \Pi} \operatorname*{argmin}_{\pi_2 \in \Pi} J_\tau(\pi_1, \pi_2, \pi_{\mathrm{ref}}),$$

*then*

$$\mathrm{KL}(\pi^\star||\pi) \leq \mathrm{KL}(\pi^\star||\pi_{\mathrm{ref}}) - \mathrm{KL}(\pi||\pi_{\mathrm{ref}})$$

*Proof.* By definition of the prox operator, we have

$$\begin{aligned}
\pi &= \operatorname*{argmax}_{\pi_1 \in \Pi} J_\tau(\pi_1, \pi, \pi_{\mathrm{ref}}) \\
&= \operatorname*{argmax}_{\pi_1 \in \Pi} \mathbb{P}(\pi_1 \succ \pi) - \tau \, \mathrm{KL}(\pi_1, \pi_{\mathrm{ref}}) \\
&= \operatorname{Prox}(\pi_{\mathrm{ref}}, \frac{1}{\tau}\mathbb{P}(\cdot \succ \pi)).
\end{aligned} \tag{6}$$

Using Corollary 1, we have for any $\pi' \in \Pi$,

$$\frac{1}{\tau}(\mathbb{P}(\pi' \succ \pi) - \mathbb{P}(\pi \succ \pi)) \leq \mathrm{KL}(\pi'||\pi_{\mathrm{ref}}) - \mathrm{KL}(\pi'||\pi) - \mathrm{KL}(\pi||\pi_{\mathrm{ref}}). \tag{7}$$

Plugging $\pi' = \pi^\star$ into the above inequality and noting that $\mathbb{P}(\pi \succ \pi) = \frac{1}{2}$, we get

$$\frac{1}{\tau}\left(\mathbb{P}(\pi^\star \succ \pi) - \frac{1}{2}\right) \leq \mathrm{KL}(\pi^\star||\pi_{\mathrm{ref}}) - \mathrm{KL}(\pi^\star||\pi) - \mathrm{KL}(\pi||\pi_{\mathrm{ref}}).$$

Since $\pi^\star$ is a Nash equilibrium and thus $\mathbb{P}(\pi^\star \succ \pi) \geq \frac{1}{2}$, the lefthand side of the above inequality is $\geq 0$. Then we have

$$\mathrm{KL}(\pi^\star||\pi) \leq \mathrm{KL}(\pi^\star||\pi_{\mathrm{ref}}) - \mathrm{KL}(\pi||\pi_{\mathrm{ref}}).$$

$\square$

Lemma 2 implies the following properties on the trajectory $\{\pi^t\}$.

**Corollary 2.** *Denote $\pi^\star$ an Nash equilibrium such that $\operatorname{supp}(\pi^\star) = \operatorname{supp}(\pi_{\mathrm{init}})$ as guaranteed by Assumption 1. Then the following holds for the trajectory $\{\pi^t\}$ produced by COMAL:*

*1. $\mathrm{KL}(\pi^\star||\pi^{t+1}) \leq \mathrm{KL}(\pi^\star||\pi^t)$ for all $t \geq 1$.*

*2. $\sum_{t=1}^\infty \mathrm{KL}(\pi^{t+1}||\pi^t) \leq \mathrm{KL}(\pi^\star||\pi_{\mathrm{init}}) < +\infty$.*

3. *For all $t \geq 1$, it holds that for $y \in \text{supp}(\pi_{\text{init}})$, $\pi^t(y) \geq c > 0$ where $c$ is some constant $c$ depends only on $\pi^\star$ and $\pi_{\text{init}}$. This also holds even for any limit point of $\{\pi^t\}$.*

*Proof.* The first item is direct from Lemma 2. The second item is also direct by applying Lemma 2 for $t \geq 1$:

$$\sum_{t=1}^{\infty} \text{KL}(\pi^{t+1}||\pi^t) \leq \sum_{t=1}^{\infty} \text{KL}(\pi^\star||\pi^t) - \text{KL}(\pi^\star||\pi^{t+1}) \leq \text{KL}(\pi^\star||\pi_{\text{init}}) \leq D < \infty.$$

Now we consider the third item. Define $D := \text{KL}(\pi^\star||\pi_{\text{init}})$ and $p_{\min} := \min_{y \in \text{supp}(\pi^\star)} \pi^\star(y)$. By Assumption 1, $p_{\min} > 0$. Then

$$\text{KL}(\pi^\star||\pi^t) \leq D \Rightarrow p_{\min} \log \frac{p_{\min}}{\pi^t(y)} \leq D, \forall y \in \text{supp}(\pi^\star)$$

$$\Rightarrow \pi^t(y) \geq \frac{p_{\min}}{\exp(D/p_{\min})}, \forall y \in \text{supp}(\pi^\star).$$

Since the above holds for all $\pi^t$, it also holds for any limit point of $\{\pi^t\}$. $\square$

Since the sequence $\{\pi^t\}$ is bounded (all lies in the simplex), it has at least one limit point $\hat{\pi}$. The next lemma shows that a limit point must be a Nash equilibrium.

**Lemma 3.** *If $\hat{\pi}$ is a limit point of $\{\pi^t\}$, then $\hat{\pi}$ is a Nash equilibrium of $J(\pi_1, \pi_2)$.*

*Proof.* By item 2 in Corollary 2, we have $\lim_{t\to\infty} \text{KL}(\pi^{t+1}||\pi^t) = 0$. This implies $\lim_{t\to\infty} \|\pi^{t+1} - \pi^t\| = 0$. As $\hat{\pi}$ is a limit point of $\{\pi^t\}$, we let $\{\pi^k : k \in \kappa\}$ be the subsequence that converges to $\hat{\pi}$. Then by Equation (6), we have

$$\lim_{k \in \kappa, k \to \infty} \pi^{k+1} = \lim_{k \in \kappa, k \to \infty} \text{Prox}(\pi^k, \frac{1}{\tau}\mathbb{P}(\cdot \succ \pi^{k+1}))$$

$$\Rightarrow \hat{\pi} = \text{Prox}(\hat{\pi}, \frac{1}{\tau}\mathbb{P}(\cdot \succ \hat{\pi})).$$

Thus $\hat{\pi}$ is a fixed point of $\text{Prox}(\pi, \frac{1}{\tau}\mathbb{P}(\cdot \succ \pi))$. Moreover, by item 3 in Corollary 2, we have $\text{supp}(\hat{\pi}) = \text{supp}(\pi_{\text{init}})$. Now consider both the max and min player running MWU initialized with $\pi^1 = \hat{\pi}$. Then we have $\pi^t = \hat{\pi}$ for all $t \geq 1$. By Equation (7), we have for any $\pi' \in \Pi$,

$$\frac{1}{\tau} \sum_{t=1}^{\infty} \left( \mathbb{P}(\pi' \succ \hat{\pi}) - \frac{1}{2} \right) \leq \text{KL}(\pi'||\hat{\pi}) < \infty,$$

where the inequality holds since $\text{supp}(\pi') \subseteq \text{supp}(\hat{\pi})$. As a result, we get

$$\mathbb{P}(\pi' \succ \hat{\pi}) \leq \frac{1}{2}, \forall \pi' \in \Pi \Leftrightarrow \mathbb{P}(\hat{\pi} \succ \pi') \geq \frac{1}{2}, \forall \pi' \in \Pi$$

Thus $\hat{\pi}$ is a Nash equilibrium of $J(\pi_1, \pi_2)$. $\square$

**Proof of Theorem 1** By Lemma 3, we know a limit point $\hat{\pi}$ is a Nash equilibrium. Then by Corollary 2, $\{\text{KL}(\hat{\pi}||\pi^t) \geq 0\}$ is a decreasing sequence. Thus $\{\text{KL}(\hat{\pi}||\pi^t)\}$ converges. Let $\{\pi^k : k \in \kappa\}$ be a subsequence that converges to $\hat{\pi}$. Then we have

$$\lim_{t\to\infty} \text{KL}(\hat{\pi}||\pi^t) = \lim_{k \in \kappa, k\to\infty} \text{KL}(\hat{\pi}||\pi^k) = \text{KL}(\hat{\pi}||\hat{\pi}) = 0.$$

Thus we have $\lim_{t\to\infty} \pi^t = \hat{\pi}$ is a Nash equilibrium. This completed the proof of Theorem 1.

### D.2 LAST-ITERATE CONVERGENCE UNDER APPROXIMATE SOLUTIONS

This section considers the case where we can not solve the regularized game $J_\tau(\pi_1, \pi_2, \pi_{\text{ref}})$ exactly but only compute an approximate solution. Specifically, we consider the following inexact COMAL

update: denote $\hat{\pi}^{t+1} = \arg\max_{\pi_1 \in \Pi} \min_{\pi_2 \in \Pi} J_\tau(\pi_1, \pi_2, \pi^t)$ the exactly solution; the algorithm updates the next iterate $\pi^{t+1}$ as an $\varepsilon_t$-approximate solution such that

$$\mathrm{KL}(\hat{\pi}^{t+1}, \pi^{t+1}) \le \varepsilon_t = O\left(\frac{1}{t^4}\right). \tag{8}$$

We note that we can compute $\pi^{t+1}$ within $\varepsilon_t$ error using $O(\log \frac{1}{\varepsilon_t}) = O(\log t)$ iterations of Algorithm 2 (Theorem 2).

We denote $\Pi^\star$ the set of Nash equilibria such that each $\pi^\star \in \Pi^\star$ has support $\mathrm{supp}(\pi^\star) = \mathrm{supp}(\pi_{\mathrm{init}})$ as guaranteed by Assumption 1. We introduce a few quantities that depend on the Nash equilibria and the initial policy.

**Definition 4.** *We define the following constants.*

1. $p_{\mathrm{sft}} := \max\{p > 0 : \forall y \in \mathrm{supp}(\pi_{\mathrm{init}}), \pi_{\mathrm{init}}(y) \ge p\}$; $D := |\mathcal{Y}| \log \frac{1}{p_{\mathrm{sft}}}$ *so that* $\mathrm{KL}(\pi\|\pi_{\mathrm{init}}) \le D$ *for all* $\pi \in \Pi$

2. $p_{\min} := \max\{p > 0 : \exists \pi^\star \in \Pi^\star, \forall y \in \mathrm{supp}(\pi_{\mathrm{init}}), \pi^\star(y) \ge p\}$; *Let* $\pi^\star \in \Pi^\star$ *be a Nash equilibrium so that* $\pi^\star(y) \ge p_{\min}$ *holds for all* $y$ *in its support.*

3. $c_1 := \frac{p_{\min}}{\exp(D+2)/p_{\min}}$ *and* $c_2 := \frac{c_1}{\exp(1/c_1)}$.

Our main result is that if each optimization problem at iteration $t$ can be solved within approximation error $\varepsilon_t \le \frac{c_1}{3t^2}$, then COMAL converges in last-iterate to a Nash equilibrium.

**Theorem 3** (COMAL with approximate regularized game solver)**.** *Assume Assumption 1 holds. If in each iteration $t \ge 1$, the returned iterate $\pi^{t+1}$ is an $\varepsilon_t$-approximate solution to $J_\tau(\pi_1, \pi_2, \pi^t)$ as defined in* (8) *with $\varepsilon_t \le \frac{c_1^2}{9t^4}$ ($c_1$ defined in Definition 4), then $\{x^t\}$ converges to a Nash equilibrium of $J(\pi_1, \pi_2)$.*

We need the following technical lemma in the proof of Theorem 3.

**Lemma 4.** *Let $\varepsilon_t \le \frac{c_1^2}{9t^4}$. Then for all $t \ge 1$,*

1. $\mathrm{KL}(\pi^\star\|\pi^{t+1}) \le \mathrm{KL}(\pi^\star\|\pi^t) - \mathrm{KL}(\pi^{t+1}\|\pi^t) + \frac{1}{t^2}$.

2. $\min_{y \in \mathrm{supp}(\pi_{\mathrm{init}})} \pi^t(y) \ge c_2$.

3. $\lim_{t \to \infty} \|\pi^{t+1} - \pi^t\| = 0$.

4. *For any Nash equilibrium $\hat{\pi} \in \Pi$ and $t \ge 1$, we have $\mathrm{KL}(\hat{\pi}\|\pi^{t+1}) \le \mathrm{KL}(\hat{\pi}\|\pi^t) + \frac{1}{t^2}$*

*Proof.* By Lemma 2, we have $\hat{\pi}^{t+1} = \mathrm{Prox}(\pi^t, \mathbb{P}(\cdot \succ \hat{\pi}^{t+1}))$ and

$$\mathrm{KL}(\pi^\star\|\hat{\pi}^{t+1}) \le \mathrm{KL}(\pi^\star\|\pi^t) - \mathrm{KL}(\hat{\pi}^{t+1}\|\pi^t). \tag{9}$$

The above implies

$$\mathrm{KL}(\pi^\star\|\pi^{t+1}) \le \mathrm{KL}(\pi^\star\|\pi^t) - \mathrm{KL}(\pi^{t+1}\|\pi^t) + \underbrace{\mathrm{KL}(\pi^\star\|\pi^{t+1}) - \mathrm{KL}(\pi^\star\|\hat{\pi}^{t+1})}_{E_1}$$
$$+ \underbrace{\mathrm{KL}(\pi^{t+1}\|\pi^t) - \mathrm{KL}(\hat{\pi}^{t+1}\|\pi^t)}_{E_2}. \tag{10}$$

Now, we use induction to prove the claim. For the base case, we define $\pi^0 := \pi^1$ and $\varepsilon_t = 0$, then

**Base Case:** $t = 0$ Since $\pi^0 = \pi^1$, we have $\mathrm{KL}(\pi^1\|\pi^0) = 0$. Then it is clear that

$$\mathrm{KL}(\pi^\star\|\pi^1) \le \mathrm{KL}(\pi^\star\|\pi^0) - \mathrm{KL}(\pi^1\|\pi^0).$$

Moreover, by Proposition 1 and $D \ge \mathrm{KL}(\pi^\star\|\pi_{\mathrm{init}})$, we have $\min_{y \in \mathrm{supp}(\pi^1)} \pi^1(y) \ge c_1 \ge c_2$.

**Induction:** $t \geq 1$   We have

$$\text{KL}(\pi^\star || \hat{\pi}^{t+1}) \leq \text{KL}(\pi^\star || \pi^t) \tag{(9)}$$

$$\leq \text{KL}(\pi^\star || \pi_{\text{init}}) + \sum_{t=1}^{t-1} \frac{1}{t^2} \qquad \text{(inductive hypothesis)}$$

$$\leq D + 2. \qquad (D \geq \text{KL}(\pi^\star || \pi_{\text{init}}))$$

Using Proposition 1, we have $\min_{y \in \text{supp}(\pi_{\text{init}})} \hat{\pi}^{t+1}(y) \geq c_1$. By $\text{KL}(\hat{\pi}^{t+1} || \pi^{t+1}) \leq \varepsilon_t \leq 1$ and Proposition 1 again, we get $\min_{y \in \text{supp}(\pi_{\text{init}})} \pi^{t+1}(y) \geq c_2 := \frac{c_1}{\exp(1/c_1)}$. Thus, both $\hat{\pi}^{t+1}$ and $\pi^{t+1}$ are bounded away from the boundary in their support. Further by $\text{KL}(\hat{\pi}^{t+1} || \pi^{t+1}) \leq \varepsilon_t$, we have

$$\sum_y \hat{\pi}^{t+1}(y) \log \frac{\hat{\pi}^{t+1}(y)}{\pi^{t+1}(y)} \leq \varepsilon_t \Rightarrow \max_y \log \frac{\hat{\pi}^{t+1}(y)}{\pi^{t+1}(y)} \leq \frac{\varepsilon_t}{c_1}.$$

As a result, we can bound

$$E_1 = \text{KL}(\pi^\star || \pi^{t+1}) - \text{KL}(\pi^\star || \hat{\pi}^{t+1})$$

$$= \sum_y \pi^\star(y) \log \frac{\hat{\pi}^{t+1}(y)}{\pi^{t+1}(y)}$$

$$\leq \max_y \log \frac{\hat{\pi}^{t+1}(y)}{\pi^{t+1}(y)}$$

$$\leq \frac{\varepsilon_t}{c_1}.$$

Moreover, we have

$$E_2 = \text{KL}(\pi^{t+1} || \pi^t) - \text{KL}(\hat{\pi}^{t+1} || \pi^t)$$

$$= \sum_y (\pi^{t+1}(y) - \hat{\pi}^{t+1}(y)) \log \frac{\pi^{t+1}(y)}{\pi^t(y)} - \text{KL}(\hat{\pi}^{t+1} || \pi^{t+1})$$

$$\leq \left\| \pi^{t+1} - \hat{\pi}^{t+1} \right\|_1 \cdot \max_y | \log \frac{\pi^{t+1}(y)}{\pi^t(y)} |$$

$$\leq \sqrt{\text{KL}(\hat{\pi}^{t+1} || \pi^{t+1})} \cdot \log \frac{1}{c_2} \qquad \text{(Pinsker's Inequality)}$$

$$\leq \frac{2\sqrt{\varepsilon_t}}{c_1}$$

Combining the above two inequalities with (10) and noting the fact that $\varepsilon_t \leq \sqrt{\varepsilon_t}$ gives

$$\text{KL}(\pi^\star || \pi^{t+1}) \leq \text{KL}(\pi^\star || \pi^t) - \text{KL}(\pi^{t+1} || \pi^t) + \frac{3\sqrt{\varepsilon_t}}{c_1}.$$

We conclude the claim since $\varepsilon_t \leq \frac{c_1^2}{9t^4}$. This completes the proof for item 1 and item 2.

For item 3, we have $\sum_{t=1}^\infty \|\pi^{t+1} - \pi^t\| \leq \sum_{t=1}^\infty \text{KL}(\pi^{t+1} || \pi^t) \leq D + 2$. Thus $\lim_{t \to \infty} \|\pi^{t+1} - \pi^t\| = 0$.

For item 4, we can use Lemma 2 and $\hat{\pi}^{t+1} = \text{Prox}(\pi^t, \mathbb{P}(\cdot \succ \hat{\pi}^{t+1}))$ to get

$$\text{KL}(\hat{\pi} || \pi^{t+1}) \leq \text{KL}(\hat{\pi} || \pi^t) - \text{KL}(\pi^{t+1} || \pi^t) + \underbrace{\text{KL}(\hat{\pi} || \pi^{t+1}) - \text{KL}(\hat{\pi} || \hat{\pi}^{t+1})}_{E_1}$$

$$+ \underbrace{\text{KL}(\pi^{t+1} || \pi^t) - \text{KL}(\hat{\pi}^{t+1} || \pi^t)}_{E_2}. \tag{11}$$

We note that $E_2 \leq \frac{2\sqrt{\varepsilon_t}}{c_1}$ has been proved in the above. For $E_1$, we have

$$E_1 = \mathrm{KL}(\hat{\pi}||\pi^{t+1}) - \mathrm{KL}(\hat{\pi}||\hat{\pi}^{t+1})$$

$$= \sum_y \hat{\pi}(y) \log \frac{\hat{\pi}^{t+1}(y)}{\pi^{t+1}(y)}$$

$$\leq \max_y \log \frac{\hat{\pi}^{t+1}(y)}{\pi^{t+1}(y)}$$

$$\leq \frac{\varepsilon_t}{c_1}.$$

Thus we have $\mathrm{KL}(\hat{\pi}||\pi^{t+1}) \leq \mathrm{KL}(\hat{\pi}||\pi^t) + \frac{1}{t^2}$ as $\varepsilon_t \leq \frac{c_1^2}{9t^4}$. $\qquad\square$

**Proof of Theorem 3**

*Proof.* Since the sequence $\{\pi^t\}$ is bounded, it has at least one limit point $\hat{\pi}$. By item 2 in Lemma 4, we know $\hat{\pi}(y) \geq c_2$ for all $y \in \mathrm{supp}(\pi_{\mathrm{init}})$. By item 3 in Lemma 4, we have $\lim_{t\to\infty} \|\pi^{t+1} - \pi^t\| = 0$. Denote $\{\pi^k : k \in \kappa\}$ a subsequence that converges to $\hat{\pi}$. Then we have

$$\hat{\pi} = \lim_{k\in\kappa,\kappa\to\infty} \pi^{k+1}$$

$$= \lim_{k\in\kappa,\kappa\to\infty} \hat{\pi}^{k+1} \qquad (\mathrm{KL}(\hat{\pi}^{k+1}, \pi^{k+1}) \leq \varepsilon_k \text{ and } \lim_{t\to\infty} \varepsilon_t = 0)$$

$$= \lim_{k\in\kappa,\kappa\to\infty} \mathrm{Prox}(\pi^k, \frac{1}{\tau}\mathbb{P}(\cdot \succ \hat{\pi}^{k+1}))$$

$$= \lim_{k\in\kappa,\kappa\to\infty} \mathrm{Prox}(\pi^{k+1}, \frac{1}{\tau}\mathbb{P}(\cdot \succ \hat{\pi}^{k+1})) \qquad (\lim_{t\to\infty} \|\pi^{t+1} - \pi^t\| = 0)$$

$$= \lim_{k\in\kappa,\kappa\to\infty} \mathrm{Prox}(\pi^{k+1}, \frac{1}{\tau}\mathbb{P}(\cdot \succ \pi^{k+1})) \qquad (\mathrm{KL}(\hat{\pi}^{k+1}, \pi^{k+1}) \leq \varepsilon_k \text{ and } \lim_{t\to\infty} \varepsilon_t = 0)$$

$$= \mathrm{Prox}(\hat{\pi}, \frac{1}{\tau}\mathbb{P}(\cdot \succ \hat{\pi})).$$

Since $\hat{\pi}$ is a fixed point of $\mathrm{Prox}(\pi, \frac{1}{\tau}\mathbb{P}(\cdot \succ \pi))$ and $\mathrm{supp}(\hat{\pi}) = \mathrm{supp}(\pi_{\mathrm{init}})$, we can use the same proof in Lemma 3 to show that $\hat{\pi}$ is a Nash equilibrium of $J(\pi_1, \pi_2)$.

Given that $\hat{\pi}$ is a Nash equilibrium of the original game, we can apply item 4 in Lemma 4 and get

$$\mathrm{KL}(\hat{\pi}||\pi^{t+1}) \leq \mathrm{KL}(\hat{\pi}||\pi^t) + \frac{1}{t^2}.$$

Now we show the sequence $\{x^t\}$ converges to $\hat{\pi}$. Fix any $\epsilon > 0$. Let $T_1 \geq 1$ such that $\sum_{t=T_1}^{\infty} \frac{1}{t^2} < \frac{\epsilon}{2}$, Since $\hat{\pi}$ is a limit point of $\{x^t\}$, there exists $T_2 \geq T_1$ such that $\mathrm{KL}(\hat{\pi}||\pi^{T_2}) \leq \frac{\epsilon}{2}$. Then for any $t \geq T_2$, we have

$$\mathrm{KL}(\hat{\pi}||\pi^{t+1}) \leq \mathrm{KL}(\hat{\pi}||\pi^{T_2}) + \sum_{t=T_2}^{\infty} \frac{1}{t^2} \leq \frac{\epsilon}{2} + \frac{\epsilon}{2} = \epsilon.$$

Since the above holds for any $\varepsilon > 0$, we know $\lim_{t\to\infty} \mathrm{KL}(\hat{\pi}||\pi^t) = 0$ and thus $\{x^t\}$ converges to $\hat{\pi}$. This completes the proof. $\qquad\square$

### D.3 AUXILIARY PROPOSTION

**Proposition 1.** *Let $\pi_1$ and $\pi_2$ be two distributions with the same support. If there exists $p, D > 0$ such that $\min_{y\in\mathrm{supp}(\pi_1)} \pi_1(y) \geq p$ and $\mathrm{KL}(\pi_1||\pi_2) \leq D$, then $\mathrm{supp}(\pi_2) = \mathrm{supp}(\pi_1)$ and*

$$\min_{y\in\mathrm{supp}(\pi_1)} \pi_2(y) \geq \frac{p}{\exp(D/p)}.$$

*Proof.* We have

$$\text{KL}(\pi_1||\pi_2) \leq D \Rightarrow p \log \frac{p}{\pi_2(y)} \leq D, \forall y \in \text{supp}(\pi_1)$$

$$\Rightarrow \pi_2(y) \geq \frac{p}{\exp(D/p)}, \forall y \in \text{supp}(\pi_1).$$

$\square$

## E   PROOF OF THEOREM 2

We show that MWU (Algorithm 2) has linear convergence to the unique Nash equilibrium of a KL-regularized zero-sum game $J(\pi_1, \pi_2, \pi_{\text{ref}})$. We denote $\mu^\star = \pi_\tau^\star$ its unique Nash equilibrium. Our proof is inspired by (Abe et al., 2024, Lemma F.1) that give linear convergence of MWU in KL-regularized game. Here, we include a simpler proof with slightly better constants for our setting for completeness.

We prove the following descent lemma, which immediately implies Theorem 2.

**Lemma 5.** *If we choose $\eta \in (0, \frac{\tau}{\tau^2 + \frac{1}{2}}]$ in MWU (Algorithm 2), then we have for every $k \geq 1$*

$$\text{KL}(\mu^\star, \mu^{k+1}) \leq \left(1 - \frac{\eta\tau}{2}\right) \text{KL}(\mu^\star, \mu^k).$$

*Proof.* We define the gradient operator $G : \Pi \to \mathbb{R}^{|\mathcal{Y}|}$ of $J(\pi_1, \pi_2)$ and the gradient operator $A : \Pi \to \mathbb{R}^{|\mathcal{Y}|}$ of the KL regularization $\text{KL}(\pi, \pi_{\text{ref}})$ as follows.

$$G(\pi) := \mathbb{P}(\cdot \succ \pi)$$

$$A(\pi) := \nabla_\pi \text{KL}(\pi, \pi_{\text{ref}}) = \log \frac{\pi(\cdot)}{\pi_{\text{ref}}(\cdot)}.$$

We define the composite operator $F = G - \tau A$. Then MWU update in Algorithm 2 is equivalent to

$$\mu^{k+1} = \text{Prox}(\mu^k, \eta F(\mu^k)).$$

Using Corollary 1, we have

$$\langle \eta F(\mu^k), \mu^\star - \mu^{k+1} \rangle \leq \text{KL}(\mu^\star||\mu^k) - \text{KL}(\mu^\star||\mu^{k+1}) - \text{KL}(\mu^{k+1}||\mu^k)$$

We focus on the left-hand side of the above inequality. Since $\mu^\star$ is a Nash equilibrium of the regularized game with gradient $F$, we have $\langle \eta F(\mu^\star), \mu^\star - \mu^{k+1} \rangle \geq 0$ and thus

$$\langle \eta F(\mu^k), \mu^\star - \mu^{k+1} \rangle$$
$$\geq \langle \eta F(\mu^k), \mu^\star - \mu^{k+1} \rangle - \langle \eta F(\mu^\star), \mu^\star - \mu^{k+1} \rangle$$
$$= \eta \underbrace{\langle G(\mu^k) - G(\mu^{k+1}), \mu^\star - \mu^{k+1} \rangle}_{\text{term}_1} + \eta\tau \underbrace{\langle A(\mu^k) - A(\mu^\star), \mu^{k+1} - \mu^\star \rangle}_{\text{term}_2}$$
$$+ \eta \underbrace{\langle G(\mu^{k+1}) - G(\mu^\star), \mu^\star - \mu^{k+1} \rangle}_{\text{term}_3 = 0}.$$

We note that $\text{term}_3 = 0$ since $G$ is the gradient of a zero-sum game:

$$\langle G(\mu^{k+1}) - G(\mu^\star), \mu^\star - \mu^{k+1} \rangle$$
$$= \mathbb{P}(\mu^\star \succ \mu^{k+1}) + \mathbb{P}(\mu^{k+1} \succ \mu^\star) - \frac{1}{2} - \frac{1}{2} = 0.$$

For $\text{term}_2$, we can apply the three-point identity for the Bregman divergence as follows:

$$\text{term}_2 = \eta\tau \langle A(\mu^k) - A(\mu^\star), \mu^{k+1} - \mu^\star \rangle$$
$$= \eta\tau \left\langle \log \frac{\mu^k}{\mu^\star}, \mu^{k+1} - \mu^\star \right\rangle$$
$$= \eta\tau \left( \text{KL}(\mu^\star||\mu^k) - \text{KL}(\mu^{k+1}||\mu^k) + \text{KL}(\mu^{k+1}||\mu^\star) \right)$$
$$\geq \eta\tau \left( \text{KL}(\mu^\star||\mu^k) - \text{KL}(\mu^{k+1}||\mu^k) \right).$$

For $\text{term}_1$, we will use the 1-Lipschitzness of $G$ and Cauchy-Swarz inequality:

$$\text{term}_1 = \eta\langle G(\mu^k) - G(\mu^{k+1}), \mu^\star - \mu^{k+1}\rangle$$
$$\geq -\eta\left(\frac{1}{2\tau}\left\|G(\mu^k) - G(\mu^{k+1})\right\|_\infty^2 + \frac{\tau}{2}\left\|\mu^\star - \mu^{k+1}\right\|_1^2\right)$$
$$\geq -\eta\left(\frac{1}{2\tau}\left\|\mu^k - \mu^{k+1}\right\|_1^2 + \frac{\tau}{2}\left\|\mu^\star - \mu^{k+1}\right\|_1^2\right) \qquad (G \text{ is 1-Lipschitz})$$
$$\geq -\frac{\eta}{2\tau}\,\text{KL}(\mu^{k+1}\|\mu^k) - \frac{\eta\tau}{2}\,\text{KL}(\mu^\star\|\mu^{k+1})$$

Combining the above gives

$$(1 - \frac{\eta\tau}{2})\,\text{KL}(\mu^\star\|\mu^{k+1}) \leq (1 - \eta\tau)\,\text{KL}(\mu^\star\|\mu^k) - (1 - \eta\tau - \frac{\eta}{2\tau})\,\text{KL}(\mu^{k+1}\|\mu^k)$$

Let $\eta \leq \frac{1}{\tau + \frac{1}{2\tau}} = \frac{\tau}{\tau^2 + \frac{1}{2}}$, then we have $1 - \eta\tau - \frac{\eta}{2\tau} \geq 0$ and thus

$$\text{KL}(\mu^\star\|\mu^{k+1}) \leq \frac{1 - \eta\tau}{1 - \frac{\eta\tau}{2}}\,\text{KL}(\mu^\star\|\mu^k) \leq \left(1 - \frac{\eta\tau}{2}\right)\text{KL}(\mu^\star\|\mu^k).$$

This completes the proof. $\qquad\square$

## F  COMPUTING THE PROX OPERATOR USING PREFERENCE LEARNING METHODS

We include additional examples showing how existing algorithms designed for RLHF and preference optimization with neural network parameters can be adapted to solve the prox operator $\text{Prox}(\pi, \eta g)$ ($\eta > 0$ is the step size). These algorithms include RL algorithms like PPO and loss-minimization algorithms like DPO, IPO, SPPO, DRO, INPO, each of which may be preferred in certain settings.

**Reinforcement Learning algorithms**  We can use the Proximal Policy Optimization (PPO) algorithm (Schulman et al., 2017) or Group-Relative Policy optimization (GRPO) (Shao et al., 2024; Guo et al., 2025) to solve $\text{Prox}(\pi, \eta g)$. Observe that

$$\text{Prox}(\pi, \eta g) = \underset{\pi'}{\text{argmax}}\{\langle \eta g, \pi'\rangle - \text{KL}(\pi'\|\pi)\}$$
$$= \underset{\pi'}{\text{argmax}}\,\mathbb{E}_{y\sim\pi'}\left[g[y] - \eta^{-1}\cdot\text{KL}(\pi'\|\pi)\right]$$

shares the same form as the objective in (5). Typically, we parameterize $\pi' = \pi_\theta$ with neural network parameters $\theta$ and optimize over $\theta$.

**Loss minimization algorithms**  Let us denote $\hat{\pi}$ the prox operator $\text{Prox}(\pi, \eta g)$, then we have

$$\hat{\pi}[y] = \frac{\pi(y)\exp(\eta g(y))}{Z} \Leftrightarrow \log\frac{\hat{\pi}(y)}{\pi(y)} - \eta g(y) + \log Z = 0,$$

where $Z = \mathbb{E}_{y\sim\pi}[\exp(\eta g(y))]$ is the partition function. We can directly compute the partition function $Z$ and thus $\hat{\pi}$ in small tabular cases. However, the partition function is hard to compute in general large-scale applications. Several works have recently proposed to solve the above equality by optimizing the corresponding $L_2$ loss.

The Self-Play Preference Optimization (SPPO) loss (Wu et al., 2024) assumes $\log Z = \frac{\eta}{2}$ and optimizes

$$\ell_{\text{SPPO}}(\theta) = \left(\log\frac{\pi_\theta(y)}{\pi(y)} - \eta g(y) - \frac{\eta}{2}\right)^2.$$

The Direct Reward Optimization (DRO) loss (Richemond et al., 2024) parameterizes both $\hat{\pi}$ and $\log Z$ with $\theta$ and $V_\phi$ respectively and optimize[4]

$$\ell_{\text{DRO}}(\theta, \phi) = \left(\log\frac{\pi_\theta(y)}{\pi(y)} - \eta g(y) - \eta V_\phi\right)^2.$$

---

[4]we modified some constants in the original DRO loss to make it consistent with our presentation. The modification has no other effects.

The REBEL loss (Gao et al., 2024) uses *differences in rewards* to eliminate the partition function $Z$ and optimize the regression loss

$$\ell_{\text{REBEL}}(\theta) = \left( \eta^{-1} \left( \log \frac{\pi_\theta(y)}{\pi(y)} - \log \frac{\pi_\theta(y')}{\pi(y')} \right) - (g(y) - g(y')) \right)^2.$$

All the above approaches can be used to solve $\text{Prox}(\pi, \eta g)$. However, directly applying them iteratively on $J(\pi_1, \pi_2)$ is equivalent to running MWU, which provably diverges. In contrast, we can apply them in Algorithm 2 and then apply our meta-algorithm COMAL to guarantee convergence to a Nash equilibrium.

**Remark 1.** *The above approaches are versatile and work well for any $g$ that can be evaluated efficiently. In particular, we should consider using them when (1) $g = r$ is a reward function and we can efficiently query $r$; (2) $g = \mathbb{P}(\cdot \mid \mu)$ is the win rate against a reference policy $\mu$, and we can efficiently sample from $\mu$ and have oracle access to $\mathbb{P}$. These two setting are popular and practical in the LLM alignment setting.*

Now we turn attention to the more specific setting where $g$ corresponds to a preference model $\mathbb{P}$ (could be a BT model or a general preference) and that we can collect a win-loss preference data set $\mathcal{D} = \{(y_w, y_l)\}$, which is standard for LLM alignment. Although the abovementioned algorithms apply, they all require estimating $g$ (the win rate) and may be inefficient in practice. In the following, we present algorithms directly working on the sampled dataset $\mathcal{D}$ without further estimation.

**Sampled loss based on the BT preference model**  Assume $g = r$ is the reward of the Bradley-Terry model, and the dataset $\{(y_w, y_l)\}$ consists of win-lose pairs of responses. Then we can solve $\text{Prox}(\pi, \eta g)$ by optimize the DPO loss (Rafailov et al., 2024) defined as

$$\ell_{\text{DPO}}((y_w, y_l); \theta) = -\log \sigma \left( \eta^{-1} \log \frac{\pi_\theta(y_w)}{\pi(y_w)} - \eta^{-1} \log \frac{\pi_\theta(y_l)}{\pi(y_l)} \right).$$

**Sampled loss for general preference**  The DPO loss inspires many other loss functions that work under even weaker assumptions on the preference model. Now, we assume a general preference model $\mathbb{P}$ over $\mathcal{Y}$ (not necessarily the BT model). We assume $g$ is the win-rate against some policy $\mu$ such that $g_\mu(y) = \mathbb{P}[y \succ \mu] := \mathbb{E}_{y' \sim \mu}[\mathbb{P}[y \succ y']]$ (think of $\mu$ as the reference policy $\pi_{\text{ref}}$ or other online policy $\pi_t$). We assume the dataset contains win-lose pairs sampled from $\mu$: $\{y_w, y_l \sim \mu\}$. We denote the preference distribution $\lambda_{\mathbb{P}}(y, y')$ as a binary distribution:

$$\lambda_{\mathbb{P}}(y, y') = \begin{cases} (y, y') \text{ w.p. } \mathbb{P}[y \succ y'] \\ (y', y) \text{ w.p. } 1 - \mathbb{P}[y \succ y'] \end{cases} \tag{12}$$

**IPO for computing** $\text{Prox}$ **for unregularized preferences**  we first show that the IPO loss could be used to solve $\pi_\theta = \text{Prox}(\pi, \eta g_\mu)$ where $g$ is the unregularized win-rate against a reference policy $\mu$ such that $g_\mu(y) = \mathbb{P}[y \succ \mu] := \mathbb{E}_{y' \sim \mu}[\mathbb{P}[y \succ y']]$. Given a dataset of win-lose pairs sampled from $\mu$: $\{y_w, y_l \sim \mu\}$, the (population) IPO loss (Azar et al., 2024) $\ell_{\text{IPO}}(\theta, \mu)$ is defined as

$$\mathbb{E}_{\substack{(y_w, y_l) \sim \mu \\ (y^+, y^-) \sim \lambda_{\mathbb{P}}(y_w, y_l)(12)}} \left[ \left( \log \frac{\pi_\theta(y^+)}{\pi_\theta(y^-)} - \log \frac{\pi(y^+)}{\pi(y^-)} - \frac{\eta}{2} \right)^2 \right]. \tag{13}$$

Azar et al. (2024) have shown that the minimizer of the $\ell_{\text{IPO}}(\theta, \mu)$ satisfies $\pi_\theta(y) \propto \pi(y) \exp\left(-\eta \mathbb{P}[y \succ \mu]\right) \Leftrightarrow \pi_\theta = \text{Prox}(\pi, \eta g_\mu)$. Thus we can compute $\text{Prox}(\pi, \eta g_\mu)$ where $g_\mu = \mathbb{P}(\cdot \succ \mu)$ by minimizing the IPO loss.

**INPO for computing** $\text{Prox}$ **for regularized preferences**  The Iterative Nash Policy Optimization (INPO) loss (Zhang et al., 2025b) is a generalization of the IPO loss to the regularized preference setting. We show that INPO could be used to compute $\text{Prox}(\mu, \eta g_\mu^\tau)$, where $g_\mu^\tau := \nabla_\pi J_\tau(\pi, \mu, \pi_{\text{ref}}) = \mathbb{P}(\cdot \succ \mu) - \tau \log \frac{\mu(\cdot)}{\pi_{\text{ref}}(\cdot)}$ is the gradient of the regularized objective (2). Given a win-loss pair data set $\{y_w, y_l \sim \mu\}$, the INPO loss $\ell_{\text{INPO}}(\pi)$ is defined as

$$\ell_{\text{INPO}}(\pi) := \mathbb{E}_{\substack{(y_w, y_l) \sim \mu \\ (y^+, y^-) \sim \lambda_{\mathbb{P}}(y_w, y_l)(12)}} \left[ \left( \log \frac{\pi(y^+)}{\pi(y^-)} - \eta\tau \log \frac{\pi_{\text{ref}}(y^+)}{\pi_{\text{ref}}(y^-)} - (1 - \eta\tau) \log \frac{\mu(y^+)}{\mu(y^-)} - \frac{\eta}{2} \right)^2 \right].$$
(14)

It has been proved that the minimizer of the INPO loss is $\text{Prox}(\mu, \eta g_\mu^\tau)$ (Zhang et al., 2025b). Thus we can use INPO in Algorithm 2 as a regularized game solver, as we show in Algorithm 4.

### F.1 COMAL INTEGRATED WITH INPO

---

**Algorithm 4:** INPO (Zhang et al., 2025b) for solving $J_\tau(\pi_1, \pi_2, \pi_{\text{ref}})$

---

**Input:** Reference policy $\pi_{\text{ref}}$, regularization $\tau > 0$, step size $\eta > 0$, number of iterations $K \geq 1$,
      preference oracle $\mathbb{P}$.
**Output:** Approximate regularized NE policy $\mu^K$
Initialize $\mu^1 \leftarrow \pi_{\text{ref}}$
**for** $k = 1, 2, \ldots, K - 1$ **do**
    Generate response pairs $\{(y_1^{(i)}, y_2^{(i)}) \sim \mu^k\}_{i=1}^n$
    Query preference oracle $\mathbb{P}$ to get preference data $\mathcal{D}_k = \{y_w^{(i)}, y_l^{(i)}\}_{i=1}^n$
    Compute $\mu^{k+1} = \arg\min_{\pi \in \Pi} \mathbb{E}_{\mathcal{D}_k} \ell_{\text{INPO}}(\pi)$ (14)
**return** $\mu^K$

---

**Practical Implementation of COMAL** We present an implementation of COMAL in Algorithm 3 using the INPO (Zhang et al., 2025b) algorithm as a subgame solver. We remark that COMAL can also be implemented using PPO or many other preference learning algorithms, as we show in Appendix F and Appendix G. Given the implementation of these existing methods, our meta-algorithm requires minimal change but achieves last-iterate convergence to a Nash equilibrium.

In practice, COMAL provides guidance for performing iterative preference optimization: the reference policy needs to be updated in order to avoid the performance upper bound imposed by a relatively weak reference policy, however, the reference policy should not be updated at each optimization step to avoid training instability.

## G MORE PRACTICAL IMPLEMENTATIONS OF COMAL

In this section, we provide more practical implementations of COMAL using iterative GRPO (Shao et al., 2024; Guo et al., 2025), the SPPO loss (Wu et al., 2024), the DRO loss (Richemond et al., 2024), and the REBEL loss (Gao et al., 2024). All these implementations demonstrate that COMAL is simple and versatile and can be integrated with many existing methods designed for preference optimization with minimal changes.

Although the SPPO, DPO, and REBEL losses are proposed in the unregularized preference setting, we have shown how to extend these losses to compute the prox operator even for KL-regularized preferences in Appendix F. Thus, we can integrate these losses for computing the prox operator in Algorithm 2 for solving the regularized game $J_\tau(\pi_1, \pi_2, \pi_{\text{ref}})$. As a result, we get the practical implementation of COMAL by using different regularized game solvers.

We omit the instruction $x \sim \rho \in \Delta(\mathcal{X})$ for notation simplicity in the following implementations. Generalization to the contextual setting is straightforward.

### G.1 PRACTICAL IMPLEMENTATION OF COMAL USING ITERATIVE GRPO (SHAO ET AL., 2024)

We observe that the iterative GRPO procedure used in DeepSeekMath (Shao et al., 2024) and DeepSeek-R1 (Guo et al., 2025) aligns closely with COMAL's design principles: iterative GPRO updates the reference policy model to the latest policy model every few steps (every 400 steps in DeepSeek-R1 (Guo et al., 2025)), and each step solves a regularized objective. To adapt iterative

GRPO to preference alignment, one simply instantiates the reward with the win-rate induced by a preference oracle $\mathbb{P}$. We include the full algorithm for completeness below in Algorithm 5. For the GRPO objective, we can use either the original objective (Shao et al., 2024) or the unbiased Dr.GRPO objective without length and std normalization (Liu et al., 2025b).

$$
\begin{aligned}
\mathcal{J}_{\mathrm{GRPO}}(\theta) = \mathbb{E}_{\{y^{(i)}\}_{i=1}^G \sim \pi_{\mathrm{old}}} \Bigg[ & \frac{1}{G} \sum_{i=1}^G \frac{1}{|y^{(i)}|} \sum_{l=1}^{|y^{(i)}|} \Bigg\{ \min\Bigg( \frac{\pi_\theta\big(y_l^{(i)} \mid y_{<l}^{(i)}\big)}{\pi_{\mathrm{old}}\big(y_l^{(i)} \mid y_{<l}^{(i)}\big)}\, \hat{A}_{i,l}, \\
& \mathrm{clip}\Bigg( \frac{\pi_\theta\big(y_l^{(i)} \mid y_{<l}^{(i)}\big)}{\pi_{\mathrm{old}}\big(y_l^{(i)} \mid y_{<l}^{(i)}\big)},\, 1-\varepsilon,\, 1+\varepsilon \Bigg) \hat{A}_{i,l} \Bigg) - \tau_t\, \mathbb{D}_{\mathrm{KL}}\big[\pi_\theta \,\|\, \pi_{\mathrm{ref}}\big] \Bigg\} \Bigg].
\end{aligned}
\tag{15}
$$

We remark that COMAL (Algorithm 1) is a meta-algorithm that can be instantiated with any algorithm that solves the regularized game in each iteration and guarantees convergence to an exact Nash equilibrium. While we focus on using Mirror Descent (Algorithm 2) for solving the regularized game and present most implementations using MD, we can also use the clairvoyant implementation of conceptual prox (Farina et al., 2022), where our convergence result (Theorem 1) still applies. Iterative GRPO for the alignment game can be seen as the clairvoyant implementation of the conceptual prox algorithm.

---

**Algorithm 5:** Practical Implementation of COMAL using iterative GRPO (Shao et al., 2024)

**Input:** Initial policy $\pi_{\mathrm{init}}$, regularization $\{\tau_t > 0\}$, number of iterations $T \geq 1$, number of inner optimization steps $\{K_t \geq 1\}$, preference oracle $\mathbb{P}$, hyperparameter $\varepsilon$.
**Output:** Optimized policy $\pi^T$
Initialize $\pi^1, \pi_\theta, \pi_{\mathrm{ref}} \leftarrow \pi_{\mathrm{init}}$
**for** $t = 1, 2, \ldots, T-1$ **do**
  reference policy $\pi_{\mathrm{ref}} \leftarrow \pi^t$
  **for** *step* $k = 1, \ldots K_t$ **do**
    Update the old policy $\pi_{\mathrm{old}} \leftarrow \pi_\theta$
    Sample $G$ responses $\{y^{(i)}\}_{i=1}^G \sim \pi_{\mathrm{old}}$
    Query preference oracle $\mathbb{P}$ to compute the reward, i.e., empirical win-rate
    $r_i := \widehat{P}[y^{(i)} \succ \pi_{\mathrm{old}}] = \frac{1}{G} \sum_{j=1}^G \mathbb{P}[y^{(i)} \succ y^{(j)}]$ for each sample $y^{(i)}$
    Compute $\hat{A}_{i,l}$ for the $l$-th token of $y^{(i)}$ through group relative advantage estimation.
    Update the policy $\pi_\theta$ by maximizing the GRPO objective (15).
  $\pi^{t+1} \leftarrow \pi_\theta$
**return** $\pi^T$

---

## G.2 COMAL INTEGRATED WITH SPPO (WU ET AL., 2024)

We present Reg-SPPO (Algorithm 6) for solving a KL-regularized game $J_\tau(\pi_1, \pi_2, \pi_{\mathrm{ref}})$, which is the instantiation of Algorithm 2 using the SPPO loss. Then, we give a practical implementation of COMAL integrated with the SPPO loss in Algorithm 7.

## G.3 COMAL INTEGRATED WITH DRO (RICHEMOND ET AL., 2024)

We present Reg-DRO (Algorithm 8) for solving a KL regularized game $J_\tau(\pi_1, \pi_2, \pi_{\mathrm{ref}})$, which is the instantiation of Algorithm 2 using the DRO loss. Then, we give a practical implementation of COMAL integrated with the DRO loss in Algorithm 9.

## G.4 COMAL INTEGRATED WITH REBEL (GAO ET AL., 2024)

We present Reg-REBEL (Algorithm 10) for solving a KL regularized game $J_\tau(\pi_1, \pi_2, \pi_{\mathrm{ref}})$, which is the instantiation of Algorithm 2 using the REBEL loss. Then, we give a practical implementation of COMAL (Algorithm 1) integrated with the REBEL loss in Algorithm 11.

---

**Algorithm 6:** Reg-SPPO: Extension of SPPO (Wu et al., 2024) for solving KL-regularized games

---

**Input:** Reference policy $\pi_{\text{ref}}$, regularization $\tau > 0$, step size $\eta > 0$, number of rounds $K \geq 1$, preference oracle $\mathbb{P}$.

**Output:** Approximate regularized Nash equilibrium policy $\mu_K$

Initialize $\mu^1 \leftarrow \pi_{\text{ref}}$

**for** $k = 1, 2, \ldots, K - 1$ **do**

    Generate responses $\{y^{(i)} \sim \mu^k\}_{i=1}^n$

    Query preference oracle $\mathbb{P}$ to annotate the win-rate $\mathbb{P}[y^{(i)} \succ y^{(j)}], \forall i, j \in [n]$

    Form dataset $\mathcal{D}_t = \{(y^{(i)}, \widehat{P}[y^{(i)} \succ \mu^k])\}_{i \in [n]}$

    Compute $\mu^{k+1} = \mu_{\theta^{k+1}}$ where

$$\theta^{k+1} = \operatorname*{argmin}_{\theta} \ell_{\text{SPPO}}(\theta) := \mathbb{E}_{(y, \widehat{P}[y \succ \mu^k]) \sim \mathcal{D}_t} \left[ \left( \log \frac{\mu_\theta(y)}{\mu^k(y)} - \eta \left( \widehat{P}[y \succ \mu^k] - \tau \log \frac{\mu^k(y)}{\pi_{\text{ref}}(y)} - \frac{1}{2} \right) \right)^2 \right]$$

**return** $\mu^K$

---

**Algorithm 7:** Practical Implementation of COMAL integrated with Reg-SPPO (Algorithm 6)

---

**Input:** Initial policy $\pi_{\text{init}}$, regularization $\{\tau_t > 0\}$, step size $\{\eta_t > 0\}$, number of iterations $T \geq 1$, number of inner optimization steps $\{K_t \geq 1\}$, preference oracle $\mathbb{P}$.

**Output:** Optimized policy $\pi^T$

Initialize $\pi^1, \pi_{\text{ref}} \leftarrow \pi_{\text{init}}$

**for** $t = 1, 2, \ldots, T - 1$ **do**

    $\pi^{t+1} \leftarrow \text{Reg-SPPO}(\pi_{\text{ref}}, \tau_t, \eta_t, K_t, \mathbb{P})$ defined in Algorithm 6

    $\pi_{\text{ref}} \leftarrow \pi^{t+1}$

**return** $\pi^T$

---

**Algorithm 8:** Reg-DRO: Extension of DRO (Richemond et al., 2024) for solving KL-regularized games

---

**Input:** Reference policy $\pi_{\text{ref}}$, regularization $\tau > 0$, step size $\eta > 0$, number of rounds $K \geq 1$, preference oracle $\mathbb{P}$.

**Output:** Approximate regularized Nash equilibrium policy $\mu_K$

Initialize $\mu^1 \leftarrow \pi_{\text{ref}}$

**for** $k = 1, 2, \ldots, K - 1$ **do**

    Generate responses $\{y^{(i)} \sim \mu^k\}_{i=1}^n$

    Query preference oracle $\mathbb{P}$ to annotate the win-rate $\mathbb{P}[y^{(i)} \succ y^{(j)}], \forall i, j \in [n]$

    Form dataset $\mathcal{D}_t = \{(y^{(i)}, \widehat{P}[y^{(i)} \succ \mu^k])\}_{i \in [n]}$

    Compute $\mu^{k+1} = \mu_{\theta^{k+1}}$ where

$$\theta^{k+1} = \operatorname*{argmin}_{\theta} \min_{\phi} \ell_{\text{DRO}}(\theta) := \mathbb{E}_{(y, \widehat{P}[y \succ \mu^k]) \sim \mathcal{D}_t} \left[ \left( \log \frac{\mu_\theta(y)}{\mu^k(y)} - \eta \left( \widehat{P}[y \succ \mu^k] - \tau \log \frac{\mu^k(y)}{\pi_{\text{ref}}(y)} \right) - \eta V_\phi \right)^2 \right]$$

**return** $\mu^K$

---

**Algorithm 9:** Practical Implementation of COMAL integrated with Reg-DRO (Algorithm 8)

---

**Input:** Initial policy $\pi_{\text{init}}$, regularization $\{\tau_t > 0\}$, step size $\{\eta_t > 0\}$, number of iterations $T \geq 1$, number of inner optimization steps $\{K_t \geq 1\}$, preference oracle $\mathbb{P}$.

**Output:** Optimized policy $\pi^T$

Initialize $\pi^1, \pi_{\text{ref}} \leftarrow \pi_{\text{init}}$

**for** $t = 1, 2, \ldots, T - 1$ **do**

    $\pi^{t+1} \leftarrow \text{Reg-DRO}(\pi_{\text{ref}}, \tau_t, \eta_t, K_t, \mathbb{P})$ defined in Algorithm 6

    $\pi_{\text{ref}} \leftarrow \pi^{t+1}$

**return** $\pi^T$

---

---

**Algorithm 10:** Reg-REBEL: Extension of REBEL (Gao et al., 2024) for solving KL-regularized games

---

**Input:** Reference policy $\pi_{\text{ref}}$, regularization $\tau > 0$, step size $\eta > 0$, number of rounds $K \geq 1$, preference oracle $\mathbb{P}$.
**Output:** Approximate regularized Nash equilibrium policy $\mu_K$
Initialize $\mu^1 \leftarrow \pi_{\text{ref}}$
**for** $k = 1, 2, \ldots, K - 1$ **do**

    Generate responses $\{y^{(i)} \sim \mu^k\}_{i=1}^n$
    Query preference oracle $\mathbb{P}$ to annotate the win-rate $\mathbb{P}[y^{(i)} \succ y^{(j)}], \forall i, j \in [n]$
    Form dataset $\mathcal{D}_t = \{(y^{(i)}, y^{(j)}, \widehat{P}[y^{(i)} \succ \mu^k], \widehat{P}[y^{(j)} \succ \mu^k])\}_{i,j \in [n]}$
    Compute $\mu^{k+1} = \mu_{\theta^{k+1}}$ where

$$\theta^{k+1} = \underset{\theta}{\arg\min}\, \ell_{\text{REBEL}}(\theta)$$

$$\ell_{\text{REBEL}}(\theta) := \mathbb{E}_{(y,y') \sim \mathcal{D}_t}\left[\left(\eta^{-1}\left(\log \frac{\mu_\theta(y)}{\mu^k(y)} - \log \frac{\mu_\theta(y')}{\mu^k(y')}\right) - \left(\widehat{P}[y \succ \mu^k] - \tau \log \frac{\mu^k(y)}{\pi_{\text{ref}}(y)} - \widehat{P}[y' \succ \mu^k] + \tau \log \frac{\mu^k(y')}{\pi_{\text{ref}}(y')}\right)\right)^2\right]$$

**return** $\mu^K$

---

**Algorithm 11:** Practical Implementation of COMAL integrated with Reg-REBEL (Algorithm 10)

---

**Input:** Initial policy $\pi_{\text{init}}$, regularization $\{\tau_t > 0\}$, step size $\{\eta_t > 0\}$, number of iterations $T \geq 1$, number of inner optimization steps $\{K_t \geq 1\}$, preference oracle $\mathbb{P}$.
**Output:** Optimized policy $\pi^T$
Initialize $\pi^1, \pi_{\text{ref}} \leftarrow \pi_{\text{init}}$
**for** $t = 1, 2, \ldots, T - 1$ **do**

    $\pi^{t+1} \leftarrow$ Reg-REBEL$(\pi_{\text{ref}}, \tau_t, \eta_t, K_t, \mathbb{P})$ defined in Algorithm 10
    $\pi_{\text{ref}} \leftarrow \pi^{t+1}$
**return** $\pi^T$

---

# H  IMPLEMENTATION OF MIRROR-PROX AND OPTIMISTIC MULTIPLICATIVE WEIGHTS UPDATE

We note that there are other algorithms that has provable last-iterate convergence to Nash equilibrium in (unregularized) zero-sum games, including the Mirror-Prox algorithm (Nemirovski, 2004) and Optimistic Multiplicative Weights Update (OMWU) algorithm (Rakhlin & Sridharan, 2013; Syrgkanis et al., 2015; Hsieh et al., 2021). We present practical implementations of these two algorithms in the context of LLM alignment for solving $J(\pi_1, \pi_2)$ (1), where we use preference optimization algorithms to solve the prox operator as shown in §3.3 and Appendix F.

We denote the gradient $g(\pi) := \mathbb{P}(\cdot \succ \pi)$.

**Mirror-Prox**    The Mirror-Prox algorithm (Nemirovski, 2004) initialized $\pi^1 = \pi_{\text{init}}$ and updates in each iteration $t \geq 1$:

$$\pi^{t+\frac{1}{2}} = \text{Prox}(\pi^t, \eta g(\pi^t))$$
$$\pi^{t+1} = \text{Prox}(\pi^t, \eta g(\pi^{t+\frac{1}{2}}))$$

We can implement Mirror-Prox using PPO/DPO/IPO/SPPO/DRO/REBEL to compute the prox operator. Specifically, we could sample from $\pi^t$ and construct a preference dataset $D_t$ and optimize certain regression loss (IPO/DRO/REBEL) to compute $\pi^{t+\frac{1}{2}} = \text{Prox}(\pi^t, \eta g(\pi^t))$. The procedure applies to the second step in each iteration. Thus in such an implementation, we require two sampling and two optimization procedures in each iteration.

**Optimistic Multiplicative Weights Update (OMWU)**    The OMWU algorithm (Rakhlin & Sridharan, 2013) is an optimistic variant of the MWU algorithm. Although MWU diverges in zero-sum games, it has been shown that OMWU has last-iterate convergence to Nash equilibrium (Wei et al.,

2021; Hsieh et al., 2021). Initialized with $\pi^1 = \pi^{\frac{1}{2}} = \pi_{\mathrm{init}}$, OMWU updates in each iteration $t \geq 1$:

$$\pi^{t+\frac{1}{2}} = \mathrm{Prox}(\pi^t, \eta g(\pi^{t-\frac{1}{2}}))$$
$$\pi^{t+1} = \mathrm{Prox}(\pi^t, \eta g(\pi^{t+\frac{1}{2}}))$$

Similarly, we can implement OMWU to solve $J(\pi_1, \pi_2)$ using preference methods to compute the prox operator as shown in §3.3. Moreover, OMWU has an equivalent update rule: initialize $\pi^1 = \pi^0 = \pi_{\mathrm{init}}$

$$\pi^{t+1} = \mathrm{Prox}(\pi^t, 2\eta g(\pi^t) - \eta g(\pi^{t-1})),$$

which requires computing only one prox operator in each iteration.

We leave a systematic evaluation of Mirror-Prox and OMWU at a large scale, including LLM alignment, to future work.

## I  SYNTHETIC EXPERIMENTS

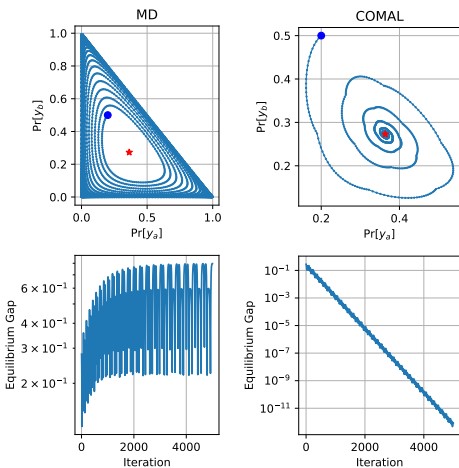

Figure 3: Dyanmics on a simple 3-dimensional preference game. The unique Nash equilibrium is $[4/11, 3/11, 4/11]$ represented as red star. We initialize all algorithms at the blue dot point $[0.2, 0.5, 0.3]$.

**Experiment Setup**  Recall that we set $\mathbb{P}[y_b \succ y_a] = \mathbb{P}[y_c \succ y_b] = 0.9$ and $\mathbb{P}[y_a \succ y_c] = 0.8$. This results in the following zero-sum game: we have policies $\Pi = \Delta(\{y_a, y_b, y_c\})$ and objective

$$J(\pi_1, \pi_2) = \pi_1^\top A \pi_2 - 0.5, \text{ where } A = \begin{bmatrix} 0.5 & 0.1 & 0.8 \\ 0.9 & 0.5 & 0.1 \\ 0.2 & 0.9 & 0.5 \end{bmatrix}.$$

The game has a unique Nash equilibrium $[4/11, 3/11, 4/11]$. We set the initial policy to be $\pi^1 = [0.2, 0.5, 0.3]$ for all algorithms. We choose $\eta = 0.3$ for iterative DPO, iterative IPO, and SPPO. We choose $\eta = 0.3$ and $\tau = 0.1$ for INPO and COMAL. For COMAL (Algorithm 3), we set $T = 200$ and $K_t = 25$ so the total number of iterations is $T \cdot K_t = 5000$.

**Experiments using noiseless gradient**  We present numerical results of mirror-descent (MD) algorithms (equivalent to MWU) and COMAL (Algorithm 1) in Figure 3. We can see that the MD algorithm diverges from the unique Nash equilibrium and suffers a large equilibrium gap, while COMAL achieves fast last-iterate convergence to the Nash equilibrium, aligned with our theoretical results (Theorem 1).

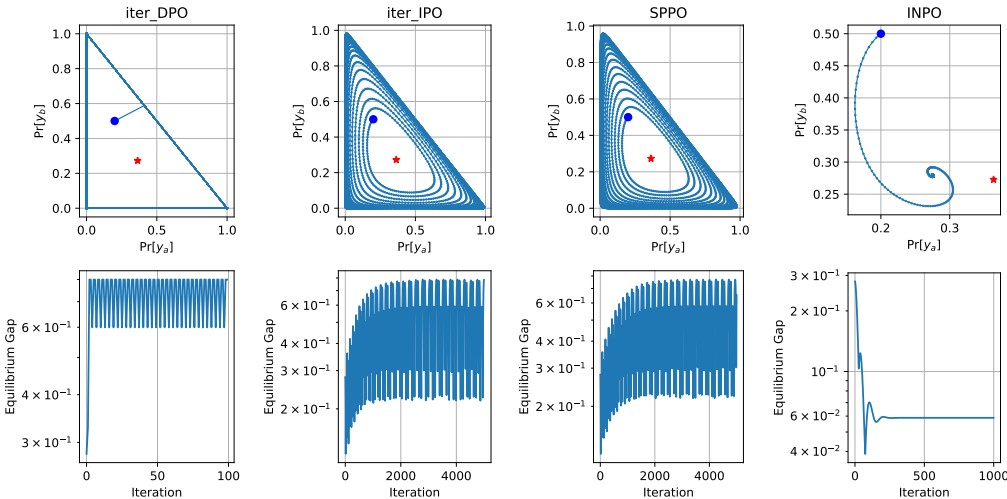

Figure 4: Dyanmics on a simple 3-dimensional preference game. The unique Nash equilibrium is $[4/11, 3/11, 4/11]$ represented as red star. We initialize all algorithms at the blue dot point $[0.2, 0.5, 0.3]$.

**Experiements using preference samples** Since the popular iterative DPO algorithm does not contain a gradient step, we also conduct experiments with only Oracle query access to the preference model. We compare the performance of various algorithms, including iterative DPO, iterative IPO, SPPO, and INPO and present results in Figure 4. The sample-only setting is also more aligned with what happens in practice. We use a sufficient number of samples in each iteration for every algorithm. As a result, the COMAL performs the same as in the noiseless gradient setting, while the iterative IPO algorithm becomes equivalent to the MD algorithm. We note the following:

*Iterative DPO:* We observe that iterative DPO cycles between extreme policies (e.g., outputting $y_a$ with probability close to 1). This is aligned with (Azar et al., 2024), where they found DPO will converge to the deterministic policy regardless of the regularization parameter in extreme preference settings. The cycling behavior of iterative DPO may be explained as follows: in each iteration, DPO converges to a nearly deterministic policy output $y$; then the new preference data shows that $y' \neq y$ is more preferred; finally, iterative DPO cycles over $\mathcal{Y}$ since the preference itself exhibits a cycle and there is no clear winner.

*Iterative IPO* (Azar et al., 2024; Calandriello et al., 2024): The IPO loss is a variant of the DPO loss, but it does not rely on the BT model assumption and works for a general preference model. However, as we have discussed before, (exactly) minimizing the IPO loss is equivalent to performing one MD step, and thus, iterative IPO is equivalent to MD up to sampling error. As a result, we observe that iterative IPO also exhibits cycling behavior.

*SPPO* (Wu et al., 2024): The SPPO algorithm (see Appendix F) is not exactly the same as MWU since SPPO assumes the partition function is always $Z = \log \frac{\eta}{2}$ which may not be the case. We observe that SPPO exhibits very similar cycling behavior as MD. We conclude that SPPO approximates MD very well in this instance and exhibits similar behavior.

*INPO* (Zhang et al., 2025b): The INPO algorithm is designed for finding the Nash equilibrium of the KL regularized game $J_\tau(\pi_1, \pi_2, \pi_{\text{ref}})$. As we proved in Theorem 2, INPO does not diverge and exhibits last-iterate convergence. However, it converges to a point that differs from the Nash equilibrium of the game $J(\pi_1, \pi_2)$ and has constant equilibrium gap.

## J HYPERPARAMETERS AND TRAINING DETAILS FOR LLM EXPERIMENTS

We follow a similar training recipe proposed in Tunstall et al. (2023) for the experiments. Specifically, at each training iteration, the models are fine-tuned for one epoch with a batch size of 32 and a maximum learning rate of $5 \times 10^{-7}$, using a cosine learning rate scheduler with 10% of warmup

steps. We conduct a grid search for the strength of the KL regularization, $\eta^{-1}$, in the loss functions of DPO, IPO and INPO, within the range of 0.001 - 0.1. INPO has another hyper-parameter $\tau$ which controls the strength of the KL regularization from the reference policy. Its value is determined following Zhang et al. (2025b), where $\eta\tau$ is set to a fixed ratio, $1/3$.

# K    LLM-BASED EXPERIMENTS WITH 1.5B LLM

In §5, we conduct experiments using an 8B LLM, Llama-3-8B-Instruct. Here, we provide additional experiments with a pre-trained smaller LLM, Qwen2-1.5B (Yang et al., 2024a). Its smaller size allows us to perform more training iterations.

## K.1    EXPERIMENTAL SETTINGS

Some of the experiment settings are identical to the settings in §5. Therefore, here we only outline the differences in the settings.

**Preference Oracle** The preference oracle we used is Llama-3-OffsetBias-8B (Park et al., 2024), which is a pairwise preference model that predicts which output is better given an instruction and a pair of outputs. Fine-tuned from Meta-Llama-3-8B-Instruct (Dubey et al., 2024), it achieves strong performance on various human preference alignment benchmarks in RewardBench (Lambert et al., 2024b). We selected it as the preference oracle for its balance of computational efficiency and alignment with human preferences, making it suitable for iterative preference optimization.

**Baselines** We include the following baselines for comparisons with COMAL: (1) SFT, which fine-tunes the pre-trained Qwen2-1.5B on the UltraChat dataset, with the resulting checkpoint serving as the starting point and/or reference policy for the other training algorithms; (2) vanilla DPO (Rafailov et al., 2024) and (3) vanilla IPO (Azar et al., 2024), where one training iteration is performed over the entire instruction set of UltraFeedback with output pairs sampled from the SFT policy; (4) INPO (Zhang et al., 2025b), where each iteration of training is performed on a single data split; (5) iterative IPO, which follows a training setting similar to INPO but without the KL regularization with respect to the reference policy.

**Evaluations** We chose to use the same preference oracle used during data generation, Llama-3-OffsetBias-8B, as the evaluator. This decision was made to maintain a controlled experimental setting, ensuring that the preference oracle the model learns to fit is also the one used to evaluate its performance.

**Training Details** We follow the training recipe proposed in Tunstall et al. (2023) for the experiments. Specifically, at each training iteration, the models are fine-tuned for 3 epochs with a batch size of 32 and maximum learning rate of $5 \times 10^{-7}$, using a linear learning rate scheduler where 10% of the steps are for warmup and the rest for linearly decreasing the rate. The checkpoints are selected based on their validation loss on the UltraFeedback dataset. The training is performed on 8 NVIDIA A6000 Ada GPUs with 48GB memory, and one training iteration over the 10K instructions takes around 5 hours. Due to the relatively high computational requirements and the large number of training iterations we tested (up to 42), we opted to use a moderately sized LLM and did not conduct an exhaustive hyper-parameter search, instead referencing settings from previous work when appropriate.

**Hyper-Parameters** We conduct a grid search for the strength of the KL regularization, $\eta^{-1}$, in both vanilla DPO and IPO. We found that DPO achieves the best performance when $\eta^{-1}$ is set to 0.01, while IPO achieves the best performance when $\eta^{-1}$ is set within the range of 0.002 - 0.01. We then choose the value of $\eta^{-1}$ to be 0.002 to encourage larger learning steps. This value of $\eta$ is also used for iterative IPO. For INPO, we compare two settings where $\eta^{-1}$ is set to 0.002 and 0.01, corresponding to a small and a large regularization respectively. INPO has another hyper-parameter $\tau$ which controls the strength of the KL regularization from the reference policy. We determine its value following the setting of Zhang et al. (2025b), where $\eta\tau$ is set to a fixed ratio, $1/3$. Regarding COMAL, which is implemented based on INPO as outlined in Algorithm 3, $\eta^{-1}$ is also set to 0.002 at the beginning of the training. The reference policy used in COMAL is updated when the first optimization step begins to converge or overfit, and $\eta^{-1}$ is increased to 0.01 to improve training stability.

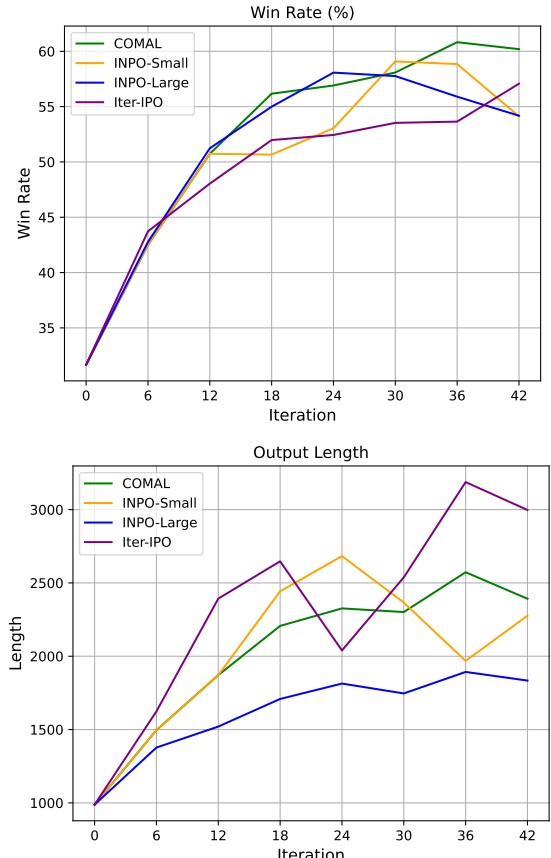

Figure 5: Comparisons of Iterative IPO (Iter-IPO), INPO, and COMAL. The average win rates of the trained checkpoints against the best checkpoints of each training algorithm, and the average lengths of the outputs are compared. For INPO, two variations with a small regularization ($\eta^{-1} = 0.002$, INPO-Small) and a large regularization ($\eta^{-1} = 0.01$, INPO-Large) are compared.

Table 5: Performance comparison of different training algorithms. The row v.s. column win rate (%) is reported. The *best* checkpoints produced by each training algorithm are compared. For INPO, we report two variations with a small regularization ($\eta^{-1} = 0.002$, INPO-Small) and a large regularization ($\eta^{-1} = 0.01$, INPO-Large).

| Row/Column | SFT | DPO | IPO | Iter-IPO | INPO-Large | INPO-Small | COMAL | Avg |
|---|---|---|---|---|---|---|---|---|
| Iter-IPO | 67.33 | 62.36 | 58.76 | 50.00 | 48.20 | 44.72 | 44.10 | 53.64 |
| INPO-Large | **77.02** | 69.81 | 67.83 | 51.80 | 50.00 | 46.21 | 44.84 | 58.22 |
| INPO-Small | 73.66 | 66.21 | 66.46 | 55.28 | 53.79 | 50.00 | 48.70 | 59.16 |
| COMAL | 74.53 | **70.56** | **68.82** | **55.90** | **55.16** | **51.30** | **50.00** | **60.90** |

## K.2 RESULT ANALYSIS

Figure 5 presents the training dynamics of three iterative preference optimization algorithms we compared: iterative IPO (Iter-IPO), INPO with a small and a large regularization (INPO-Small and INPO-Large), and COMAL, which are demonstrated by their checkpoints' win rates against the *best checkpoints* produced by 7 different algorithms: SFT, IPO, DPO, Iter-IPO, INPO-Small, INPO-Large, COMAL, and the average lengths of their outputs. For INPO and COMAL, the model is trained for up to 42 iterations, equivalent to 7 training rounds over the entire instruction set since it has been split into 6 subsets. We note that:

Table 6: Performance comparison of different training algorithms. The row v.s. column win rate (%) is reported. The *last* checkpoints produced by each training algorithm are compared. For INPO, we report two variations with a small regularization ($\eta^{-1} = 0.002$, INPO-Small) and a large regularization ($\eta^{-1} = 0.01$, INPO-Large).

| Row/Column | SFT | DPO | IPO | Iter-IPO | INPO-Large | INPO-Small | COMAL | Avg |
|---|---|---|---|---|---|---|---|---|
| Iter-IPO | 67.33 | 62.36 | 58.76 | 50.00 | 50.93 | 49.07 | 45.47 | 54.84 |
| INPO-Large | 70.43 | 62.98 | 61.61 | 49.07 | 50.00 | 48.07 | 41.61 | 54.83 |
| INPO-Small | 68.57 | 61.12 | 59.88 | 50.93 | 51.93 | 50.00 | 43.23 | 55.09 |
| COMAL | **74.53** | **67.83** | **65.09** | **54.53** | **58.39** | **56.77** | **50.00** | **61.02** |

(1) Iter-IPO shows a quicker improvement rate at the beginning of the training, but its performance begins to lag behind other algorithms after the first training round with a rapid increase in output length, which indicates the inherent instability of this training algorithm.

(2) INPO achieves stronger performance and larger improvement rates compared to Iter-IPO. However, the win rates of both INPO-Small and INPO-Large start to decrease after 5 training rounds. We suspect this suggests that INPO has started to converge and/or overfit. Moreover, for INPO-Small, its performance shows only a minor improvement and even a slight decline during training rounds 2 to 4 (iterations 12 - 24). Therefore, for COMAL, which shares the same training trajectory as INPO-Small for the first two training rounds, we update the reference policy at the beginning of the third training round, following the optimization process described in Algorithm 3.

(3) COMAL is able to further improve the model performance with the updated reference policy. Notably, its performance continues to improve up until the 6th training round, when the other algorithms begin to degrade, demonstrating the benefit of updating the reference policy.

Table 5 provides pairwise comparisons between the *best* checkpoints of the iterative preference optimization algorithms and a few baselines. It demonstrates the clear advantage of COMAL, which is able to achieve a win rate that is strictly above 50% against all the other checkpoints. The comparison of the *final* checkpoints of different algorithms after the last iteration is presented in Table 6, where COMAL is able to achieve significantly better performance thanks to its stability.

## L    LIMITATIONS

**Provable Guarantee on Computing the Prox Operators**    Our theoretical guarantee on the last-iterate convergence of COMAL relies on computing the prox operator and solving a regularized game approximately. Although we provide many practical loss minimization approaches that compute the prox operator, the applicability of our results in practical LLM settings lacks a provable guarantee since the losses could be highly non-convex, for which no provably efficient algorithms exist. We also remark that our analysis is non-trivial and novel, which gives a more robust guarantee than existing works (Perolat et al., 2021; Sokota et al., 2023; Abe et al., 2024) that require solving the regularized game *exactly*.

**Theoretical Convergence Guarantees**    Our Theorem 1 provides asymptotic last-iterate convergence to exact Nash equilibrium and Theorem 2 gives non-asymptotic $(1/\varepsilon^2)$ convergence to an $\varepsilon$-approximate Nash equilibrium when we choose the regularization $\tau = O(\varepsilon)$. Here we discuss the possibility of achieving non-asymptotic convergence with non-vanishing regularization $\tau$. We remark that there are algorithms with $\ell_2$ regularization that have polynomial last-iterate convergence rates (Cai et al., 2022). However, it is unclear whether these algorithms with $\ell_2$ regularization are practical in large-scale LLM settings, as no known efficient implementations exist. In contrast, prox operators with entropy regularization can be computed using practical preference optimization algorithms such as DPO, IPO, INPO as we discussed in §F. Regarding the possibility of establishing a last-iterate convergence rate of our algorithm, we note that a uniform convergence rate for algorithms of this type is unlikely, as suggested by a recent work (Cai et al., 2024a). Here, "uniform" refers to an upper bound on the duality gap that holds for all instances. While it may be possible to obtain a weaker instance-dependent rate, similar to the ones in (Wei et al., 2021), under a unique Nash

equilibrium assumption, the rate depends on a problem-dependent parameter that could be arbitrarily large and difficult to characterize—particularly in the LLM setting. As such, such rates offer limited practical guidance for implementation or for understanding convergence speed in realistic scenarios. Nevertheless, getting a convergence rate is an interesting question and we leave it for future work.

