# OpenReview forum: "COMAL: A Convergent Meta-Algorithm for Aligning LLMs with General Preferences"
_ICLR.cc/2026/Conference — ICLR 2026 Poster_

### Official Review · Reviewer_wCNu · 2025-10-23

**Soundness:** 3
**Presentation:** 4
**Contribution:** 2
**Rating:** 6
**Confidence:** 3

**Summary:**

This paper introduces a method of general preference optimization that is guaranteed converging to Nash Equilibrium of the preference optimization game, by simply changing to updating the reference policy. The authors have shown the previous methods suffering from convergence via synthetic examples, while their proposed method COMAL resolves this problem. The experiments are conducted on two base models, which have shown stable training curve and improved win rates.

**Strengths:**

The manuscript is well-written and every point is extremely clear and well-explained. The authors provided synthetic examples to clearly demonstrate the problems of previous methods.

This manuscript is meanwhile exhaustive. Basically all existing methods in general preference optimization have been discussed and compared. The analysis and the experiments are thorough.

The most valuable plots to me are in Fig 2, demonstrating the stability of the proposed methods and the instability of the others. The authors manage to conduct longer iterations running and manage to show stability of the training curve.

**Weaknesses:**

- The main concern is that although the difference from Magnetic Preference Optimization (MPO) [Wang et al.] has been discussed in the paper, the contribution on top of MPO feels small.

- Another issue is that results on standard benchmarks, which are more important, don't show significant improvement. I'm not sure if it is due to not using a preference model such as [3] directly. Another notable point is that INPO unlike SPPO didn't apply pairwise comparison to estimate the preference and conduct mirror descent. Anyways, I think the limited improvement weaken the manuscript.

The paper overall is very good. It's just the two concerns above bothering me. I'm happy to increase the rating if the concerns are addressed (at least to some extent).

Minor:
 - Missing Related Work [1,2]

[1] Pásztor, B., Buening, T. K., & Krause, A. Stackelberg Learning from Human Feedback: Preference Optimization as a Sequential Game. In Eighteenth European Workshop on Reinforcement Learning.
[2] Tang, X., Yoon, S., Son, S., Yuan, H., Gu, Q., & Bogunovic, I. (2025). Game-Theoretic Regularized Self-Play Alignment of Large Language Models. arXiv preprint arXiv:2503.00030.
[3] Zhang, Y., Zhang, G., Wu, Y., Xu, K., & Gu, Q. (2024). General preference modeling with preference representations for aligning language models.

**Questions:**

I agree that the original Nash Equilibrium should be the first goal of solving a game. But in practice, especially in the complex environment of preference optimization, removing regularization (or updating reference model) is better or not is unclear. At least according to INPO, keeping the regularization leads to significant improvement on standard benchmarks.

---

> ### Author Response · Authors · 2025-11-23
> **Response to Reviewer wCNu [Part 1]**
>
> We are grateful for the reviewer’s insightful comments and thoughtful review.
>
> > “The main concern is that although the difference from Magnetic Preference Optimization (MPO) [Wang et al.] has been discussed in the paper, the contribution on top of MPO feels small.”
>
> Thank you for the comment. We have provided a more detailed comparison in the updated paper:
>
> The concurrent and independent work (Wang et al., 2025) also presents a last-iterate convergent method for NLHF. Their algorithm is based on the Magnetic Mirror Descent (MMD) method (Sokato et al., 2022), which is an implementation of the conceptual prox algorithm and is the same as our work. The main differences between the two works are:
>
> 1. The theoretical results of last-iterate convergence in (Wang et al., 2025) require solving each regularized game **exactly**. This requires infinitely many iterations to solve each subgame (see their Theorems 3.4 and 3.7). In contrast, we prove last-iterate convergence under the weaker assumption that each regularized game is solved **approximately**.
>
> 2. The experiments in (Wang et al., 2025) compare only reward-based methods such as PPO and DPO, and report win rates against the base model. We conduct extensive experiments comparing COMAL with both DPO and methods for general preferences and NLHF, such as iterative IPO and INPO. Moreover, we not only report improved win rates against the base model, but also the win rates across all baseline models. Our results show that COMAL achieves a consistent > 50\% win rate against all baseline models, which is an important property of Nash equilibrium convergence.
>
> 3. (Wang et al., 2025) does not report results on standard benchmarks such as Arena-Hard or Alpaca-Eval 2, we present a comprehensive evaluation of COMAL and baseline methods on these benchmarks. Our results show the robustness of COMAL and that the alignment tax is very mild.
>
> ---
> > “Another issue is that results on standard benchmarks, which are more important, don't show significant improvement. I'm not sure if it is due to not using a preference model such as [3] directly. Another notable point is that INPO unlike SPPO didn't apply pairwise comparison to estimate the preference and conduct mirror descent. Anyways, I think the limited improvement weaken the manuscript.”
>
> Thank you for the comment. We’d like to first clarify that experimental results on pairwise win rate are more important since this is the objective of NLHF which demonstrates the importance of last-iterate convergent methods. Win rate is also the main experiment in the original NLHF works (Munos et al., 2024).
>
> We conduct experiments on standard benchmarks to demonstrate the robustness of COMAL even when the objective differs from the preference oracle used for training. Our results show that the alignment tax is very mild and COMAL achieves comparable performance to other standard alignment methods. We believe the lack of significant improvement on the standard benchmarks is due to a discrepancy between the training objective and the preference oracle (GPT-4) used on those benchmarks:
> (1) The preference distribution of GPT-4 differs from the preference oracle used in our training. As a result, Arena-Hard constitutes an out-of-distribution evaluation for our method.
> (2) Recent work [1] has shown that LLM-based evaluators can exhibit intransitive preferences. Since Arena-Hard evaluates each model by comparing it to a fixed baseline (GPT-4), this evaluation setup is not fully aligned with our optimization objective, which aims to optimize average performance across comparisons with a range of systems under diverse, intransitive preferences.
>
> That said, the point you raised about SPPO performing better than INPO on standard benchmarks is very interesting. We remark that COMAL is a meta-algorithm and can also be integrated with SPPO (we provide the implementation in the appendix). It is an interesting future direction to explore the effects of win-rate estimation, but it is beyond the scope of the current paper.
>
> References
>
> [1] Xu, Yi, et al. "Investigating Non-Transitivity in LLM-as-a-Judge." Forty-second International Conference on Machine Learning.

---

> ### Author Response · Authors · 2025-11-23
> **Response to Reviewer wCNu [Part 2]**
>
> > "Missing Related Work [1,2] [1] Pásztor, B., Buening, T. K., & Krause, A. Stackelberg Learning from Human Feedback: Preference Optimization as a Sequential Game. In Eighteenth European Workshop on Reinforcement Learning. [2] Tang, X., Yoon, S., Son, S., Yuan, H., Gu, Q., & Bogunovic, I. (2025). Game-Theoretic Regularized Self-Play Alignment of Large Language Models. arXiv preprint arXiv:2503.00030. [3] Zhang, Y., Zhang, G., Wu, Y., Xu, K., & Gu, Q. (2024)."
>
> We appreciate these pointers! We discussed them in our revised manuscript.
>
> ---
> > "I agree that the original Nash Equilibrium should be the first goal of solving a game. But in practice, especially in the complex environment of preference optimization, removing regularization (or updating reference model) is better or not is unclear. At least according to INPO, keeping the regularization leads to significant improvement on standard benchmarks."
>
> Thank you for this insightful comment! We agree that whether removing the regularization can always lead to better performance is a practical question that may depend on the specific scenarios. A specific concern regarding removing the regularization is the “alignment tax,” which motivated us to conduct the evaluations on the standard benchmarks in Table 3. The results show that COMAL maintains or achieves better performance compared to the base policies on these benchmarks, indicating that removing the regularization does not introduce a notable alignment tax. Furthermore, COMAL achieves similar performance to INPO on these benchmarks.

---

> ### Comment · Reviewer_wCNu · 2025-11-27
> **Response to Authors**
>
> I thank the authors for the detailed explanations, which have addressed my concerns. Additionally, since MPO [Wang et al.] is a concurrent work, the overlap in algorithmic ideas is understandable. Despite these similarities, I believe this submission conducts a more standard and comprehensive evaluation within the LLM alignment domain.
>
> Thus, I will increase my score to 8.

---

### Official Review · Reviewer_9tAc · 2025-10-30

**Soundness:** 2
**Presentation:** 3
**Contribution:** 2
**Rating:** 4
**Confidence:** 3

**Summary:**

This paper proposes COMAL, a meta-algorithm for aligning language models with general preferences. COMAL models the alignment problem as a two-player zero-sum game and converges to a robust Nash equilibrium policy. It integrates easily with existing methods and shows significant performance gains in experiments compared to prior methods.

**Strengths:**

1 COMAL demonstrates robust performance across different model sizes, including 8B, 7B, and 1.5B models, highlighting its scalability and practical applicability.
2 COMAL can be easily integrated into existing preference optimization algorithms with minimal changes.
3 COMAL is the first algorithm with a provable guarantee of last-iterate convergence to the unregularized Nash equilibrium.

**Weaknesses:**

1 The approach only trains on a single preference oracle. This narrow training setup raises concerns about COMAL’s ability to generalize. The Arena Hard results suggest this limitation, as COMAL underperforms Iter-IPO under GPT-4 evaluation, indicating potential overfitting to the specific training oracle. The paper would be strengthened by training on heterogeneous preference sources and demonstrating robust performance across current evaluation setting.
2 The paper strongly motivates the work by arguing that BT models cannot capture intransitive preferences, framing this as a key limitation to address. However, the experiments use a mixture of two BT models (Eq. 4) as the preference oracle without verifying whether this mixture actually exhibits the claimed intransitivity. Although synthetic experiments show that the method works on hand-crafted cyclic preferences, it remains unclear whether the preference data contains meaningful non-transitive structure. Providing evidence of intransitivity in the experimental setup or validating the method on datasets with verified cyclic preferences would better support the paper’s core claims.
3 The core contribution of the paper lies in applying the prox-method to LLM alignment via adaptive reference policy updates. While this is a significant application of an existing method, it would be helpful to clarify how this approach introduces fundamentally new ideas or significantly extends existing techniques.

**Questions:**

1 Could the authors clarify why the initial win rate in Figure 2 is not around 50% as expected? Does “iteration 0” correspond to the BASE model before training or to an updated model after the first optimization step?
2 Can the authors give detailed comparision between COMAL and the concurrent work by Wang et al[1], and clarify the advantages of COMAL in terms of alignment applications?
[1] Wang, Mingzhi, et al. "Magnetic Preference Optimization: Achieving Last-iterate Convergence for Language Model Alignment." The Thirteenth International Conference on Learning Representations.

---

> ### Author Response · Authors · 2025-11-23
> **Response to Reviewer 9tAc [Part 1]**
>
> We thank the reviewer for their helpful comments. We are grateful for the recognition of the originality of our proposed algorithm and its “scalability and practical applicability.”
>
> > “The approach only trains on a single preference oracle. This narrow training setup raises concerns about COMAL’s ability to generalize. The Arena Hard results suggest this limitation, as COMAL underperforms Iter-IPO under GPT-4 evaluation, indicating potential overfitting to the specific training oracle. The paper would be strengthened by training on heterogeneous preference sources and demonstrating robust performance across current evaluation setting.”
>
> Thank you for the comment.
>
> 1. We’d like to note that the preference oracle used in Section 5 is a mixture of two BT reward models, which mimics the diverse preferences of different annotators.  As discussed below, the preference patterns represented by this mixture cannot be captured by a single BT model. Achieving strong performance under this preference oracle requires the model to achieve high rewards from both BT models, making overfitting less likely.
>
> 2. In Section K, we include experiments using an 8B general preference model, the Llama-3-OffsetBias-8B preference model, which is different from the preference oracle used in Section 5. COMAL also exhibits a consistent >50\% win rate against all other baseline methods in this setting.
>
> 3. We’d like to note that when using GPT-4 as the evaluator (Table 3), Iter-IPO is evaluated at its *best* checkpoint since its performance deteriorates drastically after that. COMAL, on the other hand, is evaluated at the last checkpoint thanks to its convergence guarantees. While Iter-IPO performs slightly better than our method on Arena-Hard, the difference is small, and the confidence intervals of the two methods overlap. This suggests that the performance gap may not be statistically meaningful. There are two likely reasons for this small gap: (1) The preference distribution of GPT-4 differs from the preference oracle used in our training. As a result, Arena-Hard constitutes an out-of-distribution evaluation for our method. (2) Recent work [1] has shown that LLM-based evaluators can exhibit intransitive preferences. Since Arena-Hard evaluates each model by comparing it to a fixed baseline (GPT-4), this evaluation setup is not fully aligned with our optimization objective, which optimizes for average performance across comparisons against a range of systems under diverse, intransitive preferences.
>
> 4. We agree that training on more diverse, heterogeneous preference sources is an interesting next step. However, due to the increased computational complexity, we leave it for future work, as our current manuscript already includes experiments with different sources and types of preference oracles.
>
> References
>
> [1] Xu, Yi, et al. "Investigating Non-Transitivity in LLM-as-a-Judge." Forty-second International Conference on Machine Learning.

---

> ### Author Response · Authors · 2025-11-23
> **Response to Reviewer 9tAc [Part 2]**
>
> > “The paper strongly motivates the work by arguing that BT models cannot capture intransitive preferences, framing this as a key limitation to address. However, the experiments use a mixture of two BT models (Eq. 4) as the preference oracle without verifying whether this mixture actually exhibits the claimed intransitivity. Although synthetic experiments show that the method works on hand-crafted cyclic preferences, it remains unclear whether the preference data contains meaningful non-transitive structure. Providing evidence of intransitivity in the experimental setup or validating the method on datasets with verified cyclic preferences would better support the paper’s core claims.”
>
> We thank the reviewer for the suggestion.
>
> 1. We would like to note that a mixture of two BT reward models could represent complex preference patterns that exhibit intransitivity or cyclic preference. Below, we provide a concrete example: Consider three items A, B, C. The first BT model has a reward $u_1(A) = 1, u_1(B) = 2, u_1(C) = 3$ and the secnd BT model has a reward $u_2(A) = 5, u_2(B) = 3, u_2(C) = 1$.  Consider a mixture that is 60\% of the first model and 40\% of the second model. We can verify that for this mixture model, $P(A>B) \approx 0.514 > 0.5$, $P(B>C) \approx 0.514 > 0.5$, and $P(C > A) \approx 0.536 > 0.5$. Thus, the preference is cyclic as $A > B > C > A$. When the mixture uses equal 50\% weights on the two BT models, we have $P(A>B) = P(B>C) \approx 0.575 > 0.5$, $P(A>C) \approx 0.551$, which cannot be represented by a single BT model since a single BT model would imply $P(A>C) > P(A>B)$ when $P(A>B) = P(B>C) > 0.5$. Please see Appendix B.2 in our revised manuscript for a more detailed discussion.
>
>
> 2. As the reviewer mentioned, we have synthetic experiments that clearly show that COMAL works well for cyclic preferences, while other methods fail to converge to the Nash equilibrium policy.  Furthermore, we include **experiments with the Llama-3-OffsetBias-8B preference model** in Appendix K, which shows that COMAL exhibits a consistent  >50\% win rate against all other baseline methods. This preference model has complex preference patterns in its pairwise comparisons. For example, simply swapping the order of two outputs in its input prompt can lead it to make a different decision on which output is better. Along with the results on a mixture of BT reward models, these results together demonstrate that COMAL has robust performance for both reward-based preference and more complex preferences.
>
> ---
> > “The core contribution of the paper lies in applying the prox-method to LLM alignment via adaptive reference policy updates. While this is a significant application of an existing method, it would be helpful to clarify how this approach introduces fundamentally new ideas or significantly extends existing techniques.”
>
> Thank you for acknowledging that COMAL is a significant application of an existing method from optimization theory to LLM alignment. Beyond bridging theory and application with strong experimental results, our contributions are significant in three aspects:
> 1. We prove last-iterate convergence even when one solves the regularized game **approximately**, whereas all previous work requires solving it **exactly**. Our theoretical result is thus stronger and more aligned with practice.
> 2. We conduct extensive experiments that run the algorithm for 18 iterations (for 8B models) and 42 iterations (for 1.5B models) and show robust and stable performance of the last-iterate convergent COMAL method. We note that all previous work on NLHF runs experiments for only 3 iterations, which is insufficient to demonstrate NLHF's robustness and the advantage of last-iterate convergence. We view our experimental demonstration of the theoretical insights as an important contribution to the literature on game-theoretic LLM alignment.
>
> ---
> > “Could the authors clarify why the initial win rate in Figure 2 is not around 50% as expected? Does “iteration 0” correspond to the BASE model before training or to an updated model after the first optimization step?”
>
> Thank you for the question.
>
> 1. The win rate in Figure 2 is computed against multiple policies (discussed in Line 423), including the policies after training. Therefore, the initial policy has a win rate under 50\% since it underperforms the compared policies. This evaluation setting, which compares a policy against multiple policies, aligns with our training objective and follows previous work [1].
>
> 2. "Iteration 0" corresponds to the BASE model before training.
>
> References
>
> [1] Munos, Rémi, et al. "Nash learning from human feedback." Proceedings of the 41st International Conference on Machine Learning. 2024.

---

> ### Author Response · Authors · 2025-11-23
> **Response to Reviewer 9tAc [Part 3]**
>
> > “Can the authors give detailed comparision between COMAL and the concurrent work by Wang et al[1], and clarify the advantages of COMAL in terms of alignment applications? [1] Wang, Mingzhi, et al. "Magnetic Preference Optimization: Achieving Last-iterate Convergence for Language Model Alignment." The Thirteenth International Conference on Learning Representations.”
>
> Thanks for the suggestion. We have included a detailed comparison in the updated paper. We also include the comparison here for your convenience.
>
> The concurrent and independent work (Wang et al., 2025) also presents a last-iterate convergent method for NLHF. Their algorithm is based on the Magnetic Mirror Descent (MMD) method (Sokato et al., 2022), which is an implementation of the conceptual prox algorithm and is the same as our work. The main differences between the two works are:
>
> 1. The theoretical results of last-iterate convergence in (Wang et al., 2025) require solving each regularized game **exactly**. This requires infinitely many iterations for solving each subgame (see their Theorem 3.4 and Theorem 3.7). In contrast, we prove last-iterate convergence under the weaker assumption that each regularized game is solved **approximately**.
>
> 2. The experiments in (Wang et al., 2025) compare only reward-based methods such as PPO and DPO, and report win rates against the base model. We conduct extensive experiments comparing COMAL with both DPO and methods for general preferences and NLHF, such as iterative IPO and INPO. Moreover, we not only report improved win rate against the base model, but also the win rate between all baseline models. Our results show that COMAL achieves a consistent > 50\% win rate against all baseline models, which is an important property of Nash equilibrium convergence.
>
> 3. Wang et al., (2025) does not report results on standard benchmarks such as Arena-Hard or Alpaca-Eval 2, we present a comprehensive evaluation of COMAL and baseline methods on these benchmarks. Our results show the robustness of COMAL and that the alignment tax is very mild.

---

### Official Review · Reviewer_58tC · 2025-11-01

**Soundness:** 3
**Presentation:** 3
**Contribution:** 2
**Rating:** 6
**Confidence:** 4

**Summary:**

This paper proposes, a meta-algorithm designed to LLMs with general human preferences that may be intransitive or cyclic, going beyond the Bradley–Terry (BT) assumption underlying RLHF and DPO. COMAL is design for solving NLHF and seeks the Nash equilibrium (NE) policy that guarantees a 50% win rate against any competing policy. Unlike existing self-play algorithms such as iterative IPO or SPPO, which either diverge or only converge to equilibria in a regularized game, COMAL guarantees monotone KL convergence and last-iterate convergence to the NE, even under approximate subproblem solutions. Empirically, COMAL achieves the best performance among baselines (DPO, IPO, INPO, etc.) on synthetic intransitive games and real LLM fine-tuning tasks with Llama-3-8B-Instruct and Qwen2.5-7B on the UltraFeedback dataset, maintaining >60% win rate against competing algorithms and stable training dynamics.

**Strengths:**

Strengths:

1. theoretical contribution: COMAL provides a last-iterate convergence guarantee to the unregularized Nash equilibrium in the alignment game, while prior methods only achieved average-iterate or regularized convergence.

2. The experiments is comprehensive, convincingly show that the algorithms outperform existing approaches.

**Weaknesses:**

1. The main technique used to achieve last-iterate convergence is the introduction of proximal steps, which is a standard approach in optimization theory. Therefore, the theoretical contribution of this work appears limited.

2. As the authors note, Wang et al. also obtain last-iterate convergence results for the original game. I would appreciate a more comprehensive comparison between the two approaches, rather than mentioning it only in a footnote. Since these works are concurrent, I will not lower my rating due to similarity; however, for scientific completeness, a detailed comparison would be valuable.

3. Experiments: NLHF is primarily proposed to address cyclic preferences. However, the current experimental setup still follows the standard binary comparison framework and evaluates performance accordingly, leaving a conceptual gap between the theoretical motivation and the empirical validation.

The experimental design in [1] offers an interesting and informative direction. In the UltraFeedback dataset, each response is evaluated across multiple dimensions, such as honesty, truthfulness, and helpfulness. These dimensions can naturally be viewed as distinct “voters” or preference sources, making it possible to construct datasets with cyclic preferences. For example:

Cycle 1: Honest ≻ Truthful ≻ Helpful ≻ Honest

Cycle 2: IF ≻ Truthful ≻ Helpful ≻ IF

Cycle 3: IF ≻ Honest ≻ Helpful ≻ IF

Cycle 4: IF ≻ Honest ≻ Truthful ≻ IF

Exploring such cyclic-preference settings would align more closely with the theoretical foundation of NLHF and provide stronger empirical support for its claims. I suggest the authors consider such setups in future work.

[1] Y. Zhang, G. Zhang, Y. Wu, K. Xu, Q. Gu. Beyond Bradley–Terry Models: A General Preference Model for Language Model Alignment.

---
Other Related Work:

The paper emphasizes the importance of convergence to the Nash equilibrium (NE) of the original game, rather than to the modified KL-regularized game. Paper [2] provides evidence supporting this distinction. Specifically, when human preferences are cyclic (e.g., A ≻ B ≻ C ≻ A), if the preference profile contains a Condorcet winner, the NE of the original game corresponds to that winner; if not, the NE must be a mixed-strategy equilibrium.

[2] Kaizhao Liu, Qi Long, Zhekun Shi, Weijie J. Su, Jiancong Xiao. Statistical Impossibility and Possibility of Aligning LLMs with Human Preferences: From Condorcet Paradox to Nash Equilibrium.

**Questions:**

N/A

---

> ### Author Response · Authors · 2025-11-23
> **Response to Reviewer 58tC [Part 1]**
>
> We appreciate the reviewer’s insightful comments and helpful suggestions.
>
> > “The main technique used to achieve last-iterate convergence is the introduction of proximal steps, which is a standard approach in optimization theory. Therefore, the theoretical contribution of this work appears limited.”
>
> We believe our work offers a solid theoretical contribution to existing results in optimization theory. We prove last-iterate convergence even when one only solves each regularized game **approximately** (Theorem 3 in Appendix D.2). Our result is stronger and more practical than existing results that require solving the regularized game **exactly** (e.g., (Perolat et al., 2021; Abe et al., 2024; Wang et al., 2025)).
>
> ---
> > “As the authors note, Wang et al. also obtain last-iterate convergence results for the original game. I would appreciate a more comprehensive comparison between the two approaches, rather than mentioning it only in a footnote. Since these works are concurrent, I will not lower my rating due to similarity; however, for scientific completeness, a detailed comparison would be valuable.”
>
> Thanks for the suggestion. We have included a detailed comparison in the updated paper. We also include the comparison here for your convenience.
>
> The concurrent and independent work (Wang et al., 2025) also presents a last-iterate convergent method for NLHF. Their algorithm is based on the Magnetic Mirror Descent (MMD) method (Sokato et al., 2022), which is an implementation of the conceptual prox algorithm and is the same as our work. The main differences between the two works are:
>
> (a) The theoretical results of last-iterate convergence in (Wang et al., 2025) require solving each regularized game exactly. This requires an infinite number of iterations to solve each subgame (see their Theorems 3.4 and 3.7). In contrast, we prove last-iterate convergence under the weaker assumption that each regularized game is solved approximately
>
> (b) The experiments in (Wang et al., 2025) compare only reward-based methods such as PPO and DPO, while we conduct extensive experiments comparing COMAL with both DPO and methods for general preferences and NLHF, such as iterative IPO and INPO. While Wang et al. (2025) only report the win rate of their method against the base model, we report improved win rates against the base model and across all baseline methods, including DPO, IPO, and INPO. Our results show that COMAL achieves a consistent > 50% win rate against all baseline models, which is an important property of Nash equilibrium convergence.
>
> (c).(Wang et al., 2025) does not report results on standard benchmarks such as Arena-Hard or Alpaca-Eval 2, we present a comprehensive evaluation of COMAL and baseline methods on these benchmarks. Our results show the robustness of COMAL and that the alignment tax is very mild.

---

> ### Author Response · Authors · 2025-11-23
> **Response to Reviewer 58tC [Part 2]**
>
> > “Experiments: NLHF is primarily proposed to address cyclic preferences. However, the current experimental setup still follows the standard binary comparison framework and evaluates performance accordingly, leaving a conceptual gap between the theoretical motivation and the empirical validation.
> The experimental design in [1] offers an interesting and informative direction. In the UltraFeedback dataset, each response is evaluated across multiple dimensions, such as honesty, truthfulness, and helpfulness. These dimensions can naturally be viewed as distinct “voters” or preference sources, making it possible to construct datasets with cyclic preferences. For example … Exploring such cyclic-preference settings would align more closely with the theoretical foundation of NLHF and provide stronger empirical support for its claims. I suggest the authors consider such setups in future work.
> [1] Y. Zhang, G. Zhang, Y. Wu, K. Xu, Q. Gu. Beyond Bradley–Terry Models: A General Preference Model for Language Model Alignment.”
>
> 1. We’d like to clarify that our work follows precisely the NLHF approach, which is solving the two-player zero-sum game $\min_{\pi_1} \max_{\pi_2} P[\pi_1\succ \pi_2]$. The objective is defined in the general preference model, which compares two policies. We note that the preference model already models cyclic preferences: it is possible that P[A > B] > 0.5, P[B > C] > 0.5, and P[C > A] > 0.5, resulting in a cycle A > B > C > A. We also conducted experiments on synthetic data with such cyclic preferences (see Figure 1 and Section 4 for details). Our results show that COMAL is the only algorithm that converges to the Nash equilibrium of the original game, while all other methods either diverge or converge to a point far away from the Nash equilibrium.
>
> 2. We’d like to note that our evaluation setting exactly follows NLHF (please see their Table 1), which evaluates a policy by comparing it with a series of other policies. Our main evaluation (Table 1 and Table 2) follows the same approach. The evaluation is based on pairwise comparisons of policies. However, unlike the standard binary comparison framework, the policies are evaluated against multiple other policies and the average win rate is computed.
>
> 3. We remark that our results hold for any general preference model. The work [1] provides one preference model that uses latent values to improve efficiency. It is an interesting future direction to test COMAL's performance across different preference models.
>
> ---
> > “Other Related Work: The paper emphasizes the importance of convergence to the Nash equilibrium (NE) of the original game, rather than to the modified KL-regularized game. Paper [2] provides evidence supporting this distinction. Specifically, when human preferences are cyclic (e.g., A ≻ B ≻ C ≻ A), if the preference profile contains a Condorcet winner, the NE of the original game corresponds to that winner; if not, the NE must be a mixed-strategy equilibrium.
> [2] Kaizhao Liu, Qi Long, Zhekun Shi, Weijie J. Su, Jiancong Xiao. Statistical Impossibility and Possibility of Aligning LLMs with Human Preferences: From Condorcet Paradox to Nash Equilibrium”
>
> Thank you for suggesting this related work! We have added a discussion about it in the revised manuscript.

---

### Official Review · Reviewer_9AYC · 2025-11-02

**Soundness:** 2
**Presentation:** 2
**Contribution:** 2
**Rating:** 4
**Confidence:** 3

**Summary:**

This paper introduces COMAL, a meta-algorithm to align LLMs with general human preferences, which standard methods based on the Bradley-Terry (BT) model fail to capture (e.g., intransitive or cyclic preferences). It reframes alignment as a two-player zero-sum game and proposes an iterative method that, unlike existing algorithms that diverge or solve a modified problem, is theoretically guaranteed to converge in its last iterate to the exact Nash equilibrium policy. This ensures a robust 50% win rate against any competing policy. COMAL functions as an outer loop that dynamically updates a reference policy, allowing it to be integrated with many existing preference optimization methods like DPO or INPO. Experiments on synthetic games and large models like Llama-3-8B and Qwen2.5-7B demonstrate that COMAL converges correctly where others fail and consistently achieves superior win rates (e.g., over 60.2% and 56.8% against all competitors, respectively) while maintaining training stability.

**Strengths:**

- Originality: This paper proposes a meta-algorithmic framework that unifies many existing preference optimization methods (like DPO, PPO, and INPO) by re-interpreting them as practical implementations of a "Prox operator".
- Theory: This paper proves last-iterate convergence (Theorem 3) when the inner regularized game is solved only approximately.
- Clarity: This paper proposes meta-algorithm (Algorithm 1) first and then connects it to the underlying theory and practical implementations, makes the complex framework easy to follow.

**Weaknesses:**

- The main LLM experiments use a "mixture of two BT reward models" as the preference oracle, which doesn't necessarily exhibit the complex intransitivity that defines the core problem. So, experiments should include a preference oracle that explicitly has such phenomenon.
- Theorems 1 and 3 rely on solving a regularized game approximately, but it is unclear that in practice if the using the solver to solve the problem in the parameter space can produce a result which has enough close KL divergence. So, I think the theory is inadequate. Could you establish relation between the parameter approximation and distribution approximation.
- In experimental details, in LLaMA, $\eta^{-1}$ is changed from 0.002 to 0.1 after the first round, and $\eta^{-1}$ is fixed to be 0.002 in Qwen. It seems that the performance is sensitive to its choice. A more detailed discussion on the choice of $\eta$ is required, such as the ablation study. Or the reason why you adopt this schedule.

**Questions:**

See weaknesses.

---

> ### Author Response · Authors · 2025-11-23
> **Response to Reviewer 9AYC [Part 1]**
>
> We thank the reviewer for recognizing the originality of our work, and for the helpful comments.
>
> > “The main LLM experiments use a "mixture of two BT reward models" as the preference oracle, which doesn't necessarily exhibit the complex intransitivity that defines the core problem. So, experiments should include a preference oracle that explicitly has such phenomenon.”
>
> 1. We thank the reviewer for the suggestion. In fact, in our submission, we included  **experiments with the Llama-3-OffsetBias-8B preference model** in Appendix K, which shows that COMAL exhibits a consistent  >50\% win rate against all other baseline methods. This preference model has complex preference patterns in its pairwise comparisons. For example, simply swapping the order of two outputs in its input prompt can lead it to make a different decision on which output is better. Along with the results on a mixture of BT reward models, these results together demonstrate that COMAL has robust performance for both reward-based preference and more complex preferences.
>
> 2. We would like to note that a mixture of two BT reward models could represent complex preference patterns that exhibit intransitivity or cyclic preference. Below, we provide a concrete example: Consider three items A, B, C. The first BT model has a reward $u_1(A) = 1, u_1(B) = 2, u_1(C) = 3$ and the secnd BT model has a reward $u_2(A) = 5, u_2(B) = 3, u_2(C) = 1$.  Consider a mixture that is 60\% of the first model and 40\% of the second model. We can verify that for this mixture model, $P(A>B) \approx 0.514 > 0.5$, $P(B>C) \approx 0.514 > 0.5$, and $P(C > A) \approx 0.536 > 0.5$. Thus, the preference is cyclic as $A > B > C > A$. When the mixture uses equal 50\% weights on the two BT models, we have $P(A>B) = P(B>C) \approx 0.575 > 0.5$, $P(A>C) \approx 0.551$, which cannot be represented by a single BT model since a single BT model would imply $P(A>C) > P(A>B)$ when $P(A>B) = P(B>C) > 0.5$. We have updated our manuscript with this discussion in Appendix B.2.
>
> 3. We chose to use the mixture of two BT reward models as the preference oracle in our main experiment for both performance and efficiency: (a) the BT models used are stronger than similar-size preference models, making them more likely to produce stronger policies; (b) the computation required to run the two BT reward models is lower than that of running a preference model, since the latter requires pairwise comparisons among a list of candidate outputs.
>
>  ---
> > “Theorems 1 and 3 rely on solving a regularized game approximately, but it is unclear that in practice if the using the solver to solve the problem in the parameter space can produce a result which has enough close KL divergence. So, I think the theory is inadequate. Could you establish relation between the parameter approximation and distribution approximation.”
>
> We thank the reviewer for this observation.
>
> 1. We’d like to note the soundness and the robustness of our theory contribution: our method works when one can **approximately** solve a regularized game. This is already a stronger guarantee since all prior results require solving the game **exactly**, which is even less likely in practice.
>
> 2. That said, there are still gaps between theory and practice, as the expressibility of LLMs may be limited to represent all distributions. We remark that this gap between parameter and distribution approximations is a common challenge across recent game-theoretic approaches, as they all require solving a game in the distribution space. We acknowledge that theoretical results may not precisely predict practical performance. So we conduct extensive experiments and demonstrate robust practical performance of COMAL over other divergent methods or methods that only converge to regularized Nash equilibria.

---

> ### Author Response · Authors · 2025-11-23
> **Response to Reviewer 9AYC [Part 2]**
>
> > “In experimental details, in LLaMA, $eta^{-1}$ is changed from 0.002 to 0.1 after the first round, and $eta^{-1}$ is fixed to be 0.002 in Qwen. It seems that the performance is sensitive to its choice. A more detailed discussion on the choice of  $eta^{-1}$ is required, such as the ablation study. Or the reason why you adopt this schedule.”
>
> We appreciate the thoughtful comment. We have provided further details regarding the hyperparameter selection in the revised manuscript:
>
> 1. At the beginning of training, the value of $\eta^{-1}$ is determined by a grid search within the range 0.001-0.1 based on the performance of the static algorithms DPO and IPO (Lines 370-371). We found that the best performance is reached in the range 0.002-0.005 and remains relatively stable there, but starts to degrade when $\eta^{-1}$ is set to 0.001. We therefore set $\eta^{-1}$ to 0.002 at the beginning to encourage exploration.
>
> 2. After 6 training iterations, which constitute a full pass over the input instruction set, the iterative optimization algorithms (INPO and Iter-IPO) experience rapid performance degradation when training Llama-3-8B-Instruct. In contrast, training Qwen2.5-SFT remains stable. We posit that this is because Llama-3-8B-Instruct has undergone more extensive post-training, making further updates more intricate.
>
> 3. We then explored larger values of $\eta^{-1}$ for stable training and found that it must be increased to around 0.1 to maintain stability after 6 iterations. For ablation purposes, we report the performance of INPO and Iter-IPO with $\eta^{-1}$ set to both 0.002 and 0.1 in Table 1, which shows that while $\eta^{-1}=0.1$ yields more stable training, $\eta^{-1}=0.002$ provides better checkpoints before degradation begins. For COMAL, the step size $\eta^{-1}$ can be updated at each INPO training round (Algorithm 3), so we adjusted it from 0.002 to 0.1 during training. Table 1 shows that COMAL outperforms INPO and Iter-IPO with $\eta^{-1}$ set to both 0.002 and 0.1.

---

### Author Response · Authors · 2025-11-23
**General Response: Manuscript Updated**

We are grateful to all the reviewers for their constructive feedback, thoughtful comments, and insightful questions. As noted in our responses to the individual reviewers, we have updated the manuscript according to the reviews. All updated parts are marked in red for ease of review. We would greatly appreciate it if the reviewers could check our responses and the updated draft. We are happy to provide any further clarification.

---

### Meta-Review · Area_Chair_YMB3 · 2026-01-05

**Summary:**

The rebuttal meaningfully resolves the reviewer concerns : what, exactly, is the theorem buying you, and is the claimed gap vs. concurrent MPO-style work real or rhetorical? After revision, the answer is no longer hand-wavy: COMAL’s headline guarantee is last-iterate convergence to the unregularized NE under approximate subgame solves, which is strictly closer to the regime we actually inhabit with parametric LLM solvers than “assume each regularized game is solved exactly,” and this point is now stated and contrasted concretely against Wang et al. (rather than buried in a footnote). That shift justifies overruling residual “limited theoretical novelty” concerns (58tC/wCNu): proximal steps are not new, but deploying them as an outer-loop reference update that preserves last-iterate guarantees under approximation—and then demonstrating stability at long horizons (18–42 iterations, not the usual 3)—is a nontrivial contribution in this literature, where most failures are precisely last-iterate pathologies. Likewise, the “BT-mixture oracle doesn’t really test intransitivity” concern (9AYC/9tAc) is substantially neutralized: the authors both (i) provide an explicit, checkable construction showing mixtures can induce cycles / violate single-BT representability, and (ii) supplement with a non-BT preference model (OffsetBias) exhibiting richer, order-sensitive preferences, where COMAL maintains >50% win-rate against all baselines—the property the NE claim actually predicts. Finally, the standard-benchmark delta being modest (wCNu/9tAc) reads less like a flaw in the algorithm than a mismatch between training oracle and evaluation judge; the paper’s primary objective is cross-policy win-rate under general preferences, and the rebuttal makes a reasonable case that “alignment tax” is mild rather than that COMAL must dominate Arena-style evaluations. In aggregate, the rebuttal converts several “I’m uneasy” critiques into either resolved items or second-order desiderata, and the clearest signal is that wCNu explicitly states their concerns are addressed and raises their score to 8, which—paired with the strengthened theory/comparison—supports an accept decision despite remaining appetite for more heterogeneous-oracle training as future work.

**Reviewer Concerns:**

See above.

**Reviewer Scores:**

See above.

---

### Decision · Program_Chairs · 2026-01-26

Accept (Poster)